# The impact of financial development on enterprise green innovation under low carbon pilot city

Jianxiao Du[1,2], Yajie Han[1,2], Xiaoyu Cui[3]*

1 School of Accountancy, Shandong University of Finance and Economics, Jinan, China, 2 Zhuhai Fudan Innovation Institute, Hengqin, China, 3 School of Economics, Southwestern University of Finance and Economics, Chengdu, China

* 122020104003@smail.swufe.edu.cn

**Data Availability Statement:** All relevant data are within the manuscript and its Supporting Information files.

**Funding:** The author(s) received no specific funding for this work.

## Abstract

Low-carbon pilot city (LCPC) plays a pivotal role in stimulating green innovation among enterprises. However, relying solely on policy often proves less effective, necessitating support from financial development. Yet, current research frequently overlooks the impact of financial development on LCPC policy. Drawing on economic, management, and organizational psychology theories, we investigate the influence of the financial development level on enterprise green innovation in LCPC, utilizing data from listed companies between 2010 and 2018. The main finding is that LCPC facilitates institutional-level green innovation. Concurrently, financial development augments the effectiveness of LCPC policy, further expediting green innovation activities among enterprises in these pilot cities. Heterogeneity analyses reveal that financial development significantly promotes green innovation, particularly among state-owned enterprises, those with myopic management, non-high technology industries, and businesses in the southern region within LCPC. Mechanism tests identify enterprises' financing constraints and R&D investment levels as key pathways through which financial development fosters green economic development in LCPC. This study provides micro-level evidence from China elucidating the effects of environmental policies and offers practical implications for the low-carbon transformation of the manufacturing sector amid peak emissions and carbon-neutral targets. Additionally, it provides valuable guidance for other emerging economies seeking enhanced resource and environmental protection through the implementation of energy-saving and emission-reduction fiscal policy.

## 1 Introduction

In recent years, the world has been grappling with ongoing global warming and increasingly severe extreme weather events, posing a significant threat to human survival and development [1–3]. Reducing carbon emissions has emerged as a consensus within the international community [4]. The impacts stemming from global climate change transcend mere environmental concerns; they constitute a multifaceted challenge encompassing global socio-economic and

**Competing interests:** The authors have declared that no competing interests exist.

health considerations. Within this framework, the attention directed towards global per capita carbon dioxide emissions has assumed paramount importance. This is due to its direct correlation with the equilibrium of the Earth's climate system and the sustainable development of human society. As the largest developing country globally, China plays a pivotal role in global carbon emissions. According to the "China Carbon Dioxide Industry Operation Dynamics and Investment Prospects Assessment Report, 2021–2027," China ranked first globally in carbon emissions in 2020. To accomplish the objective of low-carbon development, the Chinese government initiated the LCPC policy as early as July 2010, to reach peak carbon emissions by 2030 and achieve carbon neutrality by 2060. Nevertheless, confronting the global climate challenge raises the urgent question of how to encourage enterprises to actively engage in green technological innovation. This becomes particularly pertinent when the policy of LCPC falls short in addressing the financial constraints necessary for enterprises to pursue innovation.

Within the domain of organizational psychology, contemporary profit-driven enterprises must surmount psychological inertia while enhancing psychological inclusiveness and adaptability during the implementation of organizational changes and innovation [5,6]. Considering the substantial investments, costs, risks, and unpredictable payback cycles associated with a company's decision to invest in a green technology innovation program, the motivation, and willingness to innovate within the enterprise's management team will significantly influence its readiness to undertake this pivotal initiative [7]. Enterprises will engage in green technology innovation activities actively only when well-managed, adequately funded, and with a positive outlook toward future returns [8]. Therefore, if the LCPC policy falls short in addressing the financial gap necessary for enterprise innovate, relying solely on government policy to stimulate green innovation may prove ineffective in encouraging their active participation.

Currently, enterprises primarily rely on market-based financing to address the funding gap for green innovation. Although the growth of financial institutions has enhanced service levels and reduced financing obstacles, the inherent low-risk tolerance of these institutions has hindered many enterprises from fully capitalizing on financial development. Existing studies have not definitively determined whether financial development can effectively promote green innovation in low-carbon pilot enterprises. Consequently, this study aims to clarify the interplay among financial development, LCPC policy, and carbon emissions. In addition, it seeks to explore the impact of financial development on green innovation in LCPC enterprises, along with the potential underlying mechanisms.

Based on the perspectives of organizational psychology and behavior, we have delved into the impact of the level of financial development on the green innovation of low-carbon pilot enterprises through a study of listed companies from 2010 to 2018, contributing in the following ways.

Firstly, it provides micro evidence from China for the debate on whether environmental policies and corporate green innovation are "conflicting" or "coordinating" [9]. For instance, Porter's Hypothesis suggests that the Innovation Offsets for enterprises from stringent environmental regulations can exceed the Cost of Regulation Compliance, thereby motivating enterprises to innovate [10]. Conversely, the static framework of neoclassical economics with perfectly competitive markets suggests that environmental policies are not necessarily associated with enterprises' innovations [11]. This paper empirically finds that LCPC policy will promote green innovation of enterprises using the DID method. Moreover, it discovers that the geographic region of enterprises and their ownership differences will affect the relationship between LCPC policy and green technological innovation of enterprises. This transformation of the question from "favorable or unfavorable" of LCPC policies on enterprises' green technological innovation into "when to be favorable and when to be unfavorable" helps resolve related theoretical controversies.

Finally, by introducing the level of financial development, it explains how to guide enterprises to actively carry out green innovation activities under the premise of environmental protection, expanding the theoretical boundaries of Porter's hypothesis. Previous empirical studies have often followed a simple dichotomy, solely exploring the direct effect of environmental regulation on enterprises' green innovation without examining the boundary conditions between the two. Building on the technological innovation mechanism emphasized in Porter's hypothesis, this paper finds that the level of financial development is a crucial factor influencing the green innovation of different enterprises under the LCPC policy. This provides a new perspective for explaining the effect of the policy and offers substantial empirical support for future financial development and the implementation of low-carbon policies.

The remainder of our paper is structured as follows: Section 2 provides a review of relevant literature and outlines the hypotheses to be tested. Section 3 describes the sources of data and the model setup. Section 4 presents an analysis of the regression results. Section 5 investigates heterogeneity analysis and the impact mechanism. Finally, Section 7 provides conclusions and policy recommendations.

## 2 Literature review and research hypothesis

### 2.1 Literature review

The LCPC policy has exerted a notable influence in various ways, particularly evident at the macro level [12,13]. However, there is a dearth of research at the micro level, with substantial gaps in understanding corporate innovation, especially in the realm of green innovation. On a macro scale, the LCPC policy steers the transformation and elevation of enterprises, fostering the greening of the industrial structure through the formulation and execution of a series of policy measures aimed at promoting green development [14,15]. This encompasses policy designed to encourage the adoption of clean energy [16] and the establishment of carbon emission standards [17]. The overarching goal of this policy is to propel sustainable development on an economy-wide scale, mitigating environmental burdens and enhancing resource efficiency. Nevertheless, existing research has predominantly focused on the macro-level impacts of this policy, and the specific implementation and impact mechanisms at the enterprise level remain insufficiently explored.

At the micro level, delving into the actual behaviors and innovation mechanisms of enterprises, which serve as the primary actors in policy implementation, is crucial. The tangible actions of enterprises in green innovation encompass technology research and development [18], enhancements in production processes, and product design and manufacturing [19], among other aspects. A thorough investigation into the specific actions of enterprises in green innovation sheds light on their practical strategies for navigating LCPC policy and unveils the challenges they encounter in the context of sustainable development. Moreover, the study of the innovation mechanisms crafted by enterprises in the realm of green innovation constitutes a pivotal aspect. This innovation mechanism encompasses factors such as organizational synergy and communication [20], talent development and recruitment [21], and collaborative relationships with external partners [22]. A deeper comprehension of the internal and external synergistic mechanisms forged by enterprises during the process of green innovation provides a more comprehensive understanding of how enterprises respond to LCPC policy and the ensuing impact on their overall innovation capacity.

The existing research primarily outlines the impact of LCPC policy on enterprises' green innovation, yet a comprehensive elucidation of the mechanism behind this impact remains lacking. In delving deeply into the impact of LCPC policy on enterprises' green innovation, it is imperative to progressively enhance our understanding of the underlying mechanism.

Despite the current research emphasizing the overall policy effect, the transmission and transformation mechanism of the policy within enterprises and throughout the industrial chain has not been thoroughly investigated. Our article, in particular, focuses on discerning the varied impacts of policy based on enterprise sizes, industry characteristics, and geographical environments. We also explore how these differences influence the specific mechanisms of green innovation. Moreover, while existing explanatory mechanisms predominantly concentrate on the direct impact of LCPC policy on green innovation [23,24], minimal attention has been directed toward the role of financial development in the broader green innovation ecosystem. Future research can intricately analyze the specific contributions of financial development in promoting the green innovation of enterprises. This includes examining financial support for green technology research and development, risk investment in green projects, and the incentives and support measures provided by financial institutions in the green industry chain. Such research not only enriches our comprehension of the policy influence mechanism of LCPC but also offers guidance for financial institutions to more effectively engage in green innovation.

Hence, forthcoming research should delve more profoundly into the micro level, particularly within the realm of enterprise innovation, aiming to uncover a more specific and nuanced connection between LCPC policy and financial development. This focused research endeavor is crucial for bridging the existing knowledge gap regarding the impact mechanisms of LCPC policy. Additionally, it will offer valuable insights for a more comprehensive and systematic advancement of green innovation.

## 2.2 Research hypothesis

The implementation of the LCPC policy aims to drive enterprises to reduce energy consumption and emissions in their production processes, enhance energy efficiency, and facilitate the transformation towards low-carbon industries, all in alignment with the city's goals for low-carbon development [25].

Firstly, the LCPC policy has incentive effects that promote green innovation among enterprises [26]. This policy encompasses targets for emission reduction, requirements for energy efficiency, and environmental standards that encourage enterprises to adopt environmentally friendly and sustainable production and enterprise practices [27]. The policy orientation offers distinct incentives for enterprises, thereby fueling their enthusiasm toward research, development, and innovation in the field of green technology [28]. Furthermore, this policy can influence enterprises' financial investment preferences, directing more investments toward green technology R&D to meet the requirements set by environmental policy [29].

Secondly, the LCPC policy provides policy support and resources to facilitate green innovation in enterprises [26]. The government often implements measures such as financial subsidies, tax incentives, and research funding to support enterprises aligned with the direction of green innovation, encouraging and facilitating their green innovation endeavors [30]. These policy measures reduce the risk costs of green innovation, provide enterprises with financial and resource security, and enhance their incentives to engage in green innovation activities.

Furthermore, the LCPC policy has stimulated green innovation within enterprises through market orientation and demand-pull factors [31]. Government policy guidance and environmental requirements have led to a gradual increase in demand for green products and services in the market [32]. To gain a competitive edge and meet consumer demands, enterprises need to engage in green innovation and develop products and solutions with more environmentally friendly characteristics [33].

In conclusion, the LCPC policy has stimulated green innovation in enterprises through incentives, policy support, and market demand. Based on this, we propose Hypothesis 1.

**Hypothesis 1**: LCPC policy will stimulate green innovation in enterprises.

According to organizational psychology theory, a company is a significant organization comprised of numerous subsystems that interact, depend on each other, and influence one another [34]. Consequently, during organizational change initiatives, such as green innovation, it becomes crucial to address psychological barriers among the management team [35], mitigate potential threats, overcome psychological inertia, and foster greater psychological tolerance and adaptability [5,6].

Enterprise green innovation involves high investment, costs, and risks, creating significant psychological barriers for management to undertake such initiatives and affecting managers' motivation for green innovation [36]. Without sufficient motivation, organizational change will not occur [34]. Therefore, overcoming the psychological barriers of management and motivating them has emerged as the central issue in driving green innovation within enterprises [37].

Financial development facilitates enterprises by providing them with essential investment and financial support through various mechanisms, including bank loans, venture capital, and green bonds [38]. As a result, companies can better leverage financial resources to increase their investment in green technology R&D and innovation. Furthermore, financial development also drives scientific and technological innovation and progress, offering greater technical support for enterprises' green innovation [39]. Financial institutions have accumulated extensive knowledge and expertise throughout their development and can provide consulting, assessment, and technical support services related to green technologies [40]. This technical support helps enterprises gain a better understanding of and apply green technologies, accelerating the green innovation process.

In summary, the positive influence of financial development on LCPC policy is manifested in the enhanced financial and technical support for enterprises' green innovation. Based on these observations, we propose Hypothesis 2:

**Hypothesis 2**: Financial development strengthens the positive impact of LCPC policy on enterprise green innovation.

Financial development has played a crucial role in facilitating the implementation of LCPC policy, particularly by alleviating enterprise financing constraints and thereby enhancing the policy's effectiveness [7,41].

To begin with, financial development has expanded enterprises' access to finance [42]. In the past, enterprises often encountered numerous restrictions and limitations in the financing process, including high-interest rates, strict collateral requirements, and limited financing options [43]. However, with the progress of financial development and the emergence of innovative financial institutions, a wider range of enterprise financing channels has become available, offering more diverse financing options [44]. For instance, financial institutions and instruments such as commercial banks, securities markets, venture capital, and green bond markets have provided enterprises with more convenient and flexible financing alternatives, thus reducing the constraints on enterprise financing.

Secondly, financial development has improved the cost and terms of financing for enterprises [31]. Factors such as increased competition in financial markets, innovation in financial instruments, and improved financial regulation have made it more convenient for enterprises to secure financing and obtain more favorable terms [45]. LCPC policy often necessitates enterprises to make environmental improvements and invest in green innovation, which requires substantial financial commitments [42]. Financial development has mitigated the costs and obstacles associated with enterprise financing, facilitating the implementation of LCPC policy and enabling enterprises to allocate more resources to green innovation by

providing more cost-effective access to necessary funds [46]. Simultaneously, financial resources such as credit support and government subsidies can alleviate the financial constraints faced by enterprises, thereby significantly reducing management's apprehensions regarding green technology innovation decisions and overcoming psychological barriers to such innovations.

In summary, financial development facilitates green innovation efforts under the LCPC policy by transforming the external financing landscape for enterprises, reducing their financing constraints, and inducing changes in their internal behaviors. Based on this, we propose hypothesis 3.

**Hypothesis 3**: Financial development enhances the policy effectiveness of LCPC by alleviating enterprise financing constraints.

Financial development has played a crucial role in the implementation of the LCPC policy, particularly in terms of augmenting enterprise R&D investment and further enhancing the policy's effectiveness.

Firstly, financial development has enhanced enterprises' capacity and willingness to invest in R&D. Financial development not only enables enterprises to access more capital but also improves the affordability and conditions of financing [47]. Increased competition in financial markets and the introduction of innovative financial instruments have made it easier for enterprises to secure funding at lower costs, thus reducing the financial burden associated with conducting R&D activities [48]. Moreover, financial development provides more financial products and services, such as green bonds and green investment funds, specifically designed to support low-carbon technology innovation and sustainable development projects.

Secondly, financial development provides valuable technical support and professional guidance [49]. Financial institutions have accumulated extensive technological expertise throughout their development and can offer enterprises consulting services, assessments, and technical support related to R&D activities. This technical support aids enterprises in better understanding and implementing low-carbon technologies, consequently enhancing the effectiveness and outcomes of their R&D endeavors. Furthermore, financial institutions can provide competitive financial solutions through innovative products and services tailored to technological innovation, further stimulating enterprises' R&D capabilities and innovation capacity [39].

Furthermore, financial development can mitigate the risks associated with enterprises' R&D activities through effective risk management and investment diversification [50]. Financial institutions leverage their risk assessment and pricing capabilities to provide accurate risk management advice to enterprises. This helps reduce the perceived risk and risk aversion associated with R&D activities, thus encouraging enterprises to increase their investment in low-carbon technology innovation.

In summary, financial development has played a significant role in advancing green innovation activities among enterprises by bolstering their willingness to engage in research and development, offering technical support, and facilitating risk management.

**Hypothesis 4**: Financial development contributes to the policy effect of LCPC by increasing the level of enterprise R&D investment.

## 3 Data description and model design

### 3.1 Data source

On July 19, 2010, the National Development and Reform Commission issued the Notice on the Piloting of Low-Carbon Provinces, Regions, and Cities, which identified the initial group of LCPC through a non-public selection process. Subsequently, the selection of the second and

third batches of LCPC took place in 2012 and 2017, respectively, utilizing a declaration and selection process that was more targeted compared to the previous method. Due to the limitations of available green patent data, this study focuses only on the two batches of LCPC in 2012 and 2017 to analyze the implementation effect of financial policy supporting green development. Specifically, the experimental group consists of the pilots from the second and third batches of LCPC, while the control group excludes the five provinces and eight cities from the first batch of low-carbon pilots.

This research primarily utilizes relevant indicators of listed companies and data at the prefecture-level city level spanning from 2010 to 2018. The patent data for listed companies and green patents are sourced from the State Intellectual Property Office. Other data about listed companies are obtained from the CSMAR series research database. LCPC policy data is collected from documents issued by the National Development and Reform Commission, while bank data is acquired from the financial license download data provided by the CBRC.

## 3.2 Variable selection and definition

**3.2.1 Dependent variable.**   In evaluating the quantity and quality of green innovation in LCPC, we opt to utilize the count of authorized green patents and green invention patents by publicly listed companies.

To gauge the green innovation level of enterprises, we select the total count of authorized green patents, while the count of authorized green invention patents is chosen to assess the green innovation capability and quality of enterprises. To ensure the robustness of the benchmark regression, we incorporate the counts of green patent applications and green invention patent applications as test indicators in the robustness test. This decision is made to comprehensively consider the actual performance of enterprises in the realm of green innovation.

In contrast to the total number of patent applications, the count of patent applications, the count of authorized patents provides a more comprehensive representation of enterprises' innovation prowess and levels. Simultaneously, invention patents exhibit higher technological content compared to other patent types. These two variables offer a more intuitive reflection of the output effect of enterprises' green innovation activities when compared to alternative indicators. This aligns with the overarching theme of this paper, which delves into the examination of the influence of the LCPC policy on enterprises' green technology innovation.

**3.2.2 Independent variable.**   The independent variable in our study is the LCPC policy, which is represented as a dummy variable. This variable is assigned a value of 1 for areas approved as LCPC during the policy period and 0 otherwise.

This choice is grounded in the incentive effect of the policy on enterprises' green technological innovations, and we anticipate capturing the impact of the LCPC policy on enterprises' behaviors through this variable. To evaluate the status of local financial development, we employ bank concentration as a metric. Bank concentration reflects the distribution of banks' market share in the regional market structure, highlighting disparities in the number and size of banks. Higher concentration suggests relatively weak competition, where a few banking institutions dominate market share. We utilize the Hirschman-Herfindahl index as a measure of bank concentration [51]. The specific formula for calculating this index is provided below.

$$HHI_{yt} = \sum_{i=1}^{n} \left( Y_I / Y \right)^2$$

In the formula, y represents the prefecture-level city, i represents the branch, and t represents the year. To examine the difference between high and low bank concentration, this paper uses the median as the delineation criterion. The concentration is then transformed into a dummy variable, with values greater than the median assigned a value of 1, and 0 otherwise.

**3.2.3 Control variables.** The enterprise-level control variables were selected based on the research conducted by Shao [52]. The following variables were chosen:

Age of the enterprise: The age of a company serves as an indicator of its business experience and accumulated resources. By controlling for company age, we can eliminate the influence of variations among companies of different ages on green innovation.

The proportion of the top ten shareholders: Controlling for the proportion of the top ten shareholders in a company helps mitigate the potential impact of shareholder structure on strategic decision-making and resource allocation. This enhances the reliability of research findings.

Nature of equity: The ownership nature of a company can influence its decision-making and behavior. By controlling for ownership nature, we can account for the potential effects of different ownership types on green innovation.

Debt leverage level: Companies with high levels of debt leverage may prioritize short-term profits over long-term investments in green innovation. Controlling for debt leverage level helps eliminate the influence of financial factors on green innovation.

Operating income growth rate: The revenue growth rate reflects a company's business performance and market competitiveness. By controlling for the revenue growth rate, we can address the potential impacts of differences in business capabilities on green innovation.

Total profit: Large companies with higher total profits may possess greater resources and capabilities for investing in green innovation. Controlling for total profit can mitigate potential influences of a company's financial situation on green innovation.

Table 1 presents the sources and definitions of the main variables used in this paper.

## 3.3 Model design

To verify the incentive effect of the LCPC policy on enterprises' green innovation, we have employed a DID model to verify hypothesis 1. The selection of this model is grounded in the following theoretical foundation: prior research has indicated that the LCPC policy can incentivize enterprises to lean towards the adoption of green innovation methods, aligning with sustainable production practices. Hence, we employ a DID model to compare the level of green innovation among enterprises before and after the implementation of the LCPC policy, aiming to confirm the policy's impact on green innovation.

$$Patents_{it} = \alpha + \beta_1 D1_{jt} + \beta_i controls_{jt} + \gamma_t + \delta_i + \sigma_i + \varepsilon_{it} \qquad (1)$$

**Table 1. Primary variables and explanations.**

| Variable type | Variable name | Explanations | Variable source |
|---|---|---|---|
| Dependent variable | Patents | Total number of green patent licenses | csmar |
| | Inventions | Number of green invention patents authorized | |
| Policy variables | D1 | LCPC | Development and Reform Commission |
| Financial variables | Concentration | Calculate the average by year according to the bank concentration, greater than the average takes the value of 1, otherwise, it is 0 | The data of the CBRC is calculated by itself |
| | hhi_city | The concentration of bank branches | |
| Control variable | Age | Enterprise age | csmar |
| | Ratio | The ratio of top ten shareholders (%) | |
| | Nature | Nature of equity—unique | |
| | Leverage | Debt leverage level—liabilities excluding assets | |
| | TRGR | The growth rate of enterprise gross operating income | |
| | Profit | Total profit | |

To assess the impact of the level of financial development that can enhance the effectiveness of the implementation of LCPC policy, we constructed difference-in-difference-in-difference (DDD) and quadruple-difference models to test Hypothesis 3. The rationale behind selecting these models lies in the study's observation that regions characterized by high levels of financial competition and low degrees of agglomeration tend to provide a more conducive environment for incentivizing enterprises to partake in green innovation. Consequently, we employ these difference models to ascertain whether financial development indeed exhibits a greater inclination to promote green innovation among enterprises in regions implementing LCPC policy. This approach aids in confirming the influence of financial development on environmental policy implementation.

$$Patents_{it} = \alpha + \beta_1 D1_{it} * Concentration/hhi\_city + \beta_i controls_{it} + \gamma_t + \delta_i + \sigma_i + \varepsilon_{it} \quad (2)$$

$$Patents_{it} = \alpha + \beta_1 D1_{it} * Concentration/hhi\_city * D_4 + \beta_i controls_{it} + \gamma_t + \delta_i + \sigma_i + \varepsilon_{it} \quad (3)$$

To examine the financing constraint mechanism between financial development and LCPC policy, we formulated model (4) to test hypothesis 3. The selection of this model is grounded in research indicating that financial development can alleviate enterprises' financing constraints and consequently foster green innovation. Therefore, we employ this model to investigate the impact of financial development on financing constraints and to test hypothesis 3 by assessing the influence of financing constraints on green innovation.

$$FC = \alpha + \beta_1 D1_{it} * Concentration + \beta_i controls_{it} + \gamma_t + \delta_i + \sigma_i + \varepsilon_{it} \quad (4)$$

To investigate the mechanism of R&D input share between financial development and LCPC policy, we have constructed model (5) to test hypothesis 4. The selection of this model is grounded in research indicating that financial development can enhance enterprises' levels of R&D investment, consequently fostering green innovation. Therefore, we employ this model to examine how financial development influences enterprises' R&D investment and to assess hypothesis 4 through the impact of R&D investment on green innovation.

$$R\&D = \alpha + \beta_1 D1_{it} * Concentration + \beta_i controls_{it} + \gamma_t + \delta_i + \sigma_i + \varepsilon_{it} \quad (5)$$

In these models, i represents the enterprise, and j, and t represent the city and time, respectively. γ represents the individual fixed effect, δ represents the time-fixed effect, σ represents the province fixed effect, and μ represents the random disturbance term. These models enable us to delve into and validate our research hypotheses, providing a robust methodological foundation for our study.

## 4 Results and analysis

### 4.1 Descriptive statistics

Table 2 presents descriptive statistics for key variables. The results reveal that the average number of patent applications (Patents) is 16.213, ranging from a minimum of 0 to a maximum of 781. There is a notable variation in the number of green patents across different companies, indicating clear disparities. Moving to financial variables, the median value for the concentration of bank branches (hhi_city) is 0.092, ranging from 0 to 1. The average value for the virtual financial variable (Concentration) is 0.351. Regarding policy variables associated with LCPC, the mean value is 0.425, close to 0.5, implying that approximately half of the sampled companies fall within the scope of the policy pilot. The data size discrepancy between the two control groups is small, which facilitates a better assessment of policy implementation effectiveness.

**Table 2. Descriptive statistics.**

| Variable type | Variable name | Sample size | mean | standard deviation | Min. | Max. |
|---|---|---|---|---|---|---|
| Dependent variable | Patents | 10282 | 1.583 | 16.213 | 0 | 781 |
| | Inventions | 10282 | 0.737 | 12.133 | 0 | 664 |
| Policy variables | D1 | 10282 | 0.425 | 0.494 | 0 | 1 |
| Financial variables | Concentration | 10282 | 0.351 | 0.477 | 0 | 1 |
| | hhi_ city | 10282 | 0.092 | 0.035 | 0 | 1 |
| control variable | Age | 10282 | 15.750 | 5.352 | 1 | 37 |
| | Ratio | 10282 | 58.779 | 15.206 | 10.370 | 98.585 |
| | Nature | 10282 | 1.666 | 0.589 | 1 | 4 |
| | Leverage | 10282 | 0.435 | 0.292 | 0 | 12.127 |
| | TRGR | 10282 | 0.264 | 4.017 | -0.971 | 367.532 |
| | Profit | 10282 | 15.753 | 10.759 | -23.4 | 26.618 |

## 4.2 Analysis of baseline regression results

**4.2.1 The implementation effect of LCPC policy.** Upon completion of data preparation, this paper employs DDD and multiple fixed effects models to test hypothesis 1. The regression results of LCPC policy on the number of green innovations of enterprises are presented in Table 3. Column 1 displays the regression outcome for the total number of patents granted, indicating the association between initial LCPC policy and enterprise green innovation. Subsequently, columns 2 to 6 gradually enterprise control variables.

Based on the results in column 6, the estimated coefficient, after controlling for variables and fixed effects, is 1.762, significant at the 5% level. This suggests that the number of patents granted by companies implementing LCPC increases by 1.762 compared to non-pilot regions. Similarly, Table 4 showcases the regression results of LCPC policy on the quality of green innovation exhibited by enterprises. In column 6, after accounting for control variables and fixed effects, the estimated coefficient is 1.149, significant at the 5% level. This indicates that the quality of green innovation among enterprises implementing LCPC policy is 1.051 higher than that observed in non-pilot areas. Consequently, it can be concluded that LCPC policy has the potential to stimulate green innovation among enterprises. This study examines the relationship between LCPC and green innovation, providing support for hypothesis 1.

**4.2.2 The Role of Finance in policy promotion.** Using the DDD method, we examined the impact of fiscal policy on LCPC. The results are presented in Tables 5 and 6. Table 5 displays the regression results, incorporating the interaction term between discrete bank concentration and LCPC policy. Columns (1) and (3) represent the results without any control variables, while columns (2) and (4) represent the results with control variables. The findings reveal that in LCPC, the experimental group with high bank concentration has 81.7% fewer green patent authorizations and 66.7% fewer green invention patent authorizations compared to the experimental group with low concentration. To further validate our findings, we conducted an additional analysis using continuous bank branch concentration, as shown in Table 6. Irrespective of the inclusion of control variables, the results consistently indicate that higher bank branch concentration negatively affects local enterprises' green innovation in LCPC. Conversely, lower bank concentration, indicating stronger financial competitiveness, positively influences policy implementation and facilitates innovation in supported enterprises. In essence, regions with stronger financial development exhibit more effective policy implementation, resulting in a greater quantity and quality of innovation in enterprises. These results are statistically significant at the 1% level, signifying that financial development in LCPC promotes green innovation in enterprises. Consequently, we examine the

**Table 3. Regression results of the total number of green patents and LCPC policy.**

|  | Patents | | | | | |
|---|---|---|---|---|---|---|
|  | (1) | (2) | (3) | (4) | (5) | (6) |
| D1 | 1.647** | 1.658** | 1.658** | 1.658** | 1.761** | 1.762** |
|  | (2.23) | (2.24) | (2.24) | (2.24) | (2.28) | (2.28) |
| Ratio |  | 0.018 | 0.018 | 0.018 | 0.025* | 0.025* |
|  |  | (1.32) | (1.33) | (1.33) | (1.70) | (1.71) |
| Leverage |  |  | -0.066 | -0.068 | -0.330 | -0.329 |
|  |  |  | (-0.29) | (-0.30) | (-0.79) | (-0.79) |
| TRGR |  |  |  | -0.002 | 0.002 | 0.004 |
|  |  |  |  | (-0.46) | (0.80) | (1.28) |
| Lnprofit |  |  |  |  | -0.000 | -0.000 |
|  |  |  |  |  | (-1.56) | (-1.56) |
| Nature |  |  |  |  |  | -0.397** |
|  |  |  |  |  |  | (-2.11) |
| Year fixed effect | yes | yes | yes | yes | yes | yes |
| Individual fixed effect | yes | yes | yes | yes | yes | yes |
| Provincial fixed effect | yes | yes | yes | yes | yes | yes |
| _ cons | -0.986 | -2.130 | -2.083 | -2.081 | -2.169 | -4.011 |
|  | (-0.99) | (-1.33) | (-1.38) | (-1.38) | (-1.41) | (-1.63) |
| N | 10282 | 10282 | 10282 | 10282 | 10282 | 10282 |
| $R^2$_a | 0.009 | 0.009 | 0.009 | 0.009 | 0.058 | 0.004 |

Note: The t value of the regression coefficient is shown in brackets, and

*, **, and ***represent the significance level of 10%, 5%, and 1% respectively.

relationship between financial development and green innovation in LCPC, thereby validating hypothesis 2.

## 4.3 Robustness test

**4.3.1 Robustness test of regression results.** In this study, we conducted a robustness analysis of the regression results using three different approaches: substituting the dependent variable with a proxy variable, altering the policy implementation batches, and introducing policy lag terms. The first approach involved replacing the patent authorization quantity with the patent application quantity and substituting the proxy for financial development level with venture capital finance. The results, shown in columns (1) and (4) of Table 7, validated the total green patent authorizations and total green invention patent authorizations, respectively. Furthermore, in Table 8, we presented the results of substituting the financial development level. Columns (1) and (2) demonstrated the validation for total patent authorizations and invention patent authorizations, respectively. These results consistently showed a statistically significant positive relationship within the 5% confidence interval, providing further support for the findings. The second approach focused solely on the second batch of LCPC implemented in 2012. The validation results are displayed in columns (2) and (5) of the table. The third approach involved introducing policy lag terms, which are shown in columns (3) and (6) of the table. Across all approaches, the results consistently indicated a negative relationship at a significance level of 10% or higher, unenterprising the robustness of the regression results. Consequently, the findings regarding the impact of financial development on green innovation in enterprises exhibit strong robustness.

**Table 4. Regression results of green invention patents and LCPC policy.**

| | Inventions | | | | | |
|---|---|---|---|---|---|---|
| | (1) | (2) | (3) | (4) | (5) | (6) |
| D1 | 1.051** | 1.061** | 1.060** | 1.060** | 1.149** | 1.149** |
| | (2.08) | (2.09) | (2.09) | (2.09) | (2.08) | (2.08) |
| Ratio | | 0.015 | 0.015 | 0.015 | 0.021* | 0.021* |
| | | (1.50) | (1.49) | (1.49) | (1.78) | (1.79) |
| Leverage | | | -0.069 | -0.070 | -0.294 | -0.293 |
| | | | (-0.42) | (-0.42) | (-0.84) | (-0.83) |
| TRGR | | | | -0.001 | 0.003 | 0.004 |
| | | | | (-0.34) | (1.21) | (1.40) |
| Lnprofit | | | | | -0.000 | -0.000 |
| | | | | | (-1.44) | (-1.44) |
| Nature | | | | | | -0.212 |
| | | | | | | (-1.60) |
| Year fixed effect | yes | yes | yes | yes | yes | yes |
| Individual fixed effect | yes | yes | yes | yes | yes | yes |
| Provincial fixed effect | yes | yes | yes | yes | yes | yes |
| _ cons | -0.750 | -1.690 | -1.641 | -1.640 | -1.716 | -1.380 |
| | (-1.01) | (-1.45) | (-1.49) | (-1.49) | (-1.51) | (-1.38) |
| N | 10282 | 10282 | 10282 | 10282 | 10282 | 10282 |
| $R^2$_a | 0.005 | 0.005 | 0.005 | 0.005 | 0.061 | 0.061 |

Note: The t value of the regression coefficient is shown in brackets, and
*, **, and ***represent the significance level of 10%, 5%, and 1% respectively.

**Table 5. Impact of finance on policy implementation.**

| | Patents | | Inventions | |
|---|---|---|---|---|
| | (1) | (2) | (3) | (4) |
| Interactive item | -0.817** | -0.956*** | -0.667* | -0.787** |
| | (-2.48) | (-3.10) | (-1.72) | (-2.20) |
| D1 | 1.469 | 1.529* | 1.033 | 1.082 |
| | (1.57) | (1.82) | (1.39) | (1.65) |
| Concentration | -0.724 | -0.638 | -0.615 | -0.540 |
| | (-1.20) | (-1.37) | (-1.25) | (-1.44) |
| Control variable | yes | yes | yes | yes |
| Year fixed effect | yes | yes | yes | yes |
| Individual fixed effect | yes | yes | yes | yes |
| Provincial fixed effect | yes | yes | yes | yes |
| _ cons | -0.361 | -3.211 | -0.212 | -1.857* |
| | (-0.40) | (-1.63) | (-0.43) | (-1.68) |
| N | 10282 | 10282 | 10282 | 10282 |
| $R^2$_a | 0.010 | 0.059 | 0.006 | 0.062 |

Note: The t value of the regression coefficient is shown in brackets, and
*, **, and ***represent the significance level of 10%, 5%, and 1% respectively.

**Table 6. Impact of finance on policy implementation.**

| | Patents | | Inventions | |
|---|---|---|---|---|
| | **(1)** | **(2)** | **(3)** | **(4)** |
| Interactive item | -27.038** | -30.992*** | -17.342** | -20.785*** |
| | (-2.61) | (-2.98) | (-2.52) | (-2.92) |
| D1 | 3.663** | 4.026** | 2.418* | 2.730** |
| | (2.26) | (2.62) | (1.98) | (2.41) |
| hhi_ city | -11.667 | -8.927 | -7.135 | -4.841 |
| | (-1.41) | (-1.39) | (-1.31) | (-1.36) |
| Control variable | yes | yes | yes | yes |
| Year fixed effect | yes | yes | yes | yes |
| Individual fixed effect | yes | yes | yes | yes |
| Provincial fixed effect | yes | yes | yes | yes |
| _ cons | 0.509 | -2.657 | 0.169 | -1.898 |
| | (0.45) | (-1.14) | (0.35) | (-1.48) |
| N | 10282 | 10282 | 10282 | 10282 |
| $R^2$_a | 0.009 | 0.058 | 0.005 | 0.061 |

Note: The t value of the regression coefficient is shown in brackets, and

*, **, and ***represent the significance level of 10%, 5%, and 1% respectively.

**4.3.2 Re-verification of results under the four-fold difference.** Based on the above results, it is evident that the implementation of LCPC policy significantly promotes green innovation in enterprises, and financial development also fosters research and development innovation. However, it raises the question of whether the impact of financial development on patent R&D promotion differs between green patents and other patents under the LCPC policy. In other words, does financial development exert a stronger influence on green patents

**Table 7. Robustness test of regression results.**

| | Patents | | | | Inventions | |
|---|---|---|---|---|---|---|
| | **(1)** | **(2)** | **(3)** | **(4)** | **(5)** | **(6)** |
| Interactive item | -0.755* | -0.968* | -0.510* | -0.803* | -0.880* | -0.377** |
| | (-1.71) | (-1.96) | (-1.70) | (-1.73) | (-1.70) | (-2.03) |
| D1 | 1.188** | 2.314 | 1.440* | 0.627 | 1.673*** | 1.012* |
| | (2.59) | (1.57) | (1.96) | (0.94) | (3.75) | (1.68) |
| Concentration | -0.805*** | -0.887 | -0.868 | -1.730*** | -0.661* | -0.782 |
| | (-2.90) | (-1.59) | (-1.27) | (-2.80) | (-1.77) | (-1.35) |
| control variable | yes | yes | yes | yes | Yes | yes |
| Year fixed effect | yes | yes | yes | yes | yes | yes |
| Individual fixed effect | yes | yes | yes | yes | yes | yes |
| Provincial fixed effect | Yes | yes | yes | yes | yes | yes |
| _ cons | -3.477 | -0.734 | -2.781 | -0.694 | 0.007 | -1.552 |
| | (-1.15) | (-0.56) | (-1.29) | (-0.49) | (0.00) | (-1.24) |
| N | 10282 | 7181 | 8546 | 10282 | 7181 | 8546 |
| $R^2$_a | 0.008 | 0.118 | 0.055 | 0.004 | -0.098 | 0.063 |

Note: The t value of the regression coefficient is shown in brackets, and

*, **, and ***represent the significance level of 10%, 5%, and 1% respectively.

**Table 8. Robustness test of regression results.**

|  | (1) | (2) |
|---|---|---|
|  | **Patents** | **Inventions** |
| Interactive item | 0.098** | 0.084** |
|  | (2.12) | (2.29) |
| 1.D1 | 0.473 | 0.220 |
|  | (0.91) | (0.70) |
| control variable | yes | yes |
| Year fixed effect | yes | yes |
| Individual fixed effect | yes | yes |
| Provincial fixed effect | yes | yes |
| _ cons | -2.264 | -1.878 |
|  | (-1.15) | (-1.25) |
| N | 8622 | 8622 |
| R2_a | 0.085 | 0.082 |

Note: The t value of the regression coefficient is shown in brackets, and

*, **, and ***represent the significance level of 10%, 5%, and 1% respectively.

compared to other patents in regions implementing the LCPC policy? To examine this, the study utilizes data on all patents granted to listed companies and separates the dataset into green patents and other patents. A dummy variable, D4, is introduced and defined as 1 when the patent data represents the number of green patents and 0 otherwise. High-dimensional fixed-effects regression analysis is conducted by interacting the dummy variable with dummy variables representing LCPC and financial development, and the results are presented in Table 9.

Based on the results in columns (1) and (3), preliminary conclusions can be drawn without considering any control variables. However, more accurate findings are provided in columns (2) and (4), which include control variables. Analyzing the regression results of the interaction term of the dummy variables in columns (1) and (2) reveals that financial development indeed has a greater promoting effect on the growth of green patents in regions implementing the LCPC policy. The quantity of green patents experiences a faster growth rate compared to other

**Table 9. The unique role of financial development in green innovation.**

|  | Total number of patent licenses | |
|---|---|---|
|  | (1) | (2) |
| Third-order interaction term | -54.302*** | -54.302*** |
|  | (-6.01) | (-6.01) |
| The original term and the second-order interaction term | yes | yes |
| control variable | no | yes |
| Year fixed effect | yes | yes |
| Provincial fixed effect | yes | yes |
| Corporate fixed effect | yes | yes |
| N | 9756 | 9756 |
| $R^2$_a | 0.293 | 0.294 |

Note: The t value of the regression coefficient is shown in brackets, and

*, **, and ***represent the significance level of 10%, 5%, and 1% respectively.

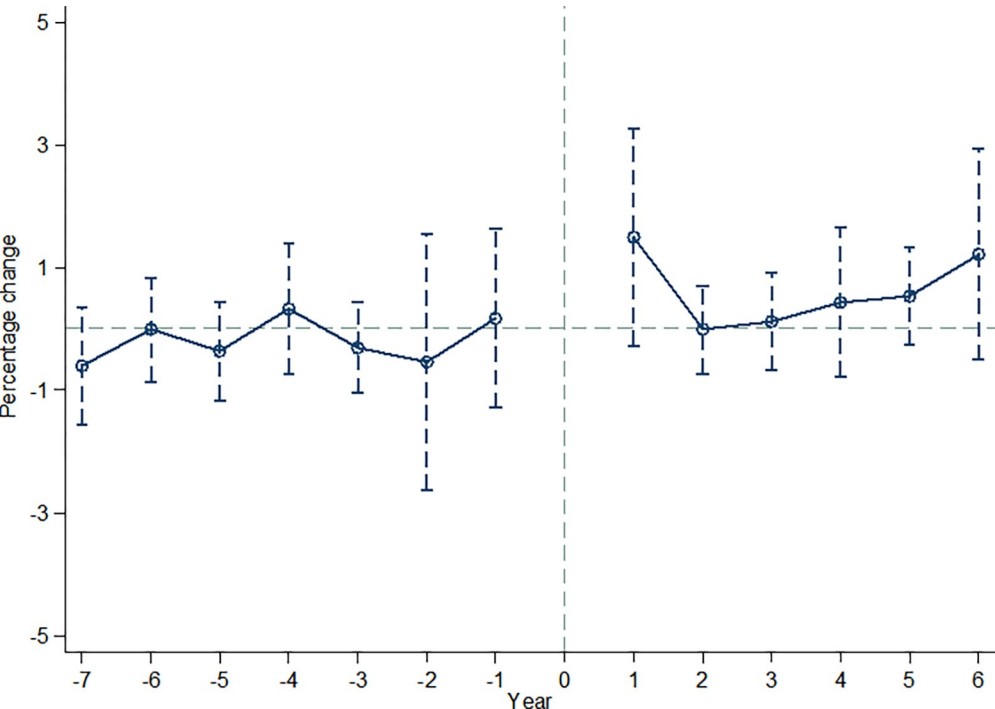

**Fig 1. Parallel trend test.**

patents, suggesting that financial development plays a vital supporting role in the implementation of urban environmental policy. Additionally, the results in columns (3) and (4) validate the significant facilitating effect of financial development on the quality of environmental policy implementation, further unenterprising the earlier hypotheses.

**4.3.3 Parallel trend test.** The effectiveness of the DDD method for policy spanning multiple periods relies on the assumption of parallel trends. This assumption suggests that the data from the treatment group and control group should exhibit similar trends before policy implementation. To examine this, we conducted a parallel trends test comparing the treatment group (policy = 1) with the control group (policy = 0). The results of the estimated model shown in Fig 1, where the horizontal axis represents time, and the vertical axis represents the average number of green patent authorizations. The vertical dashed line represents the policy implementation period, dividing the periods under examination. From the graph, it is evident that before policy implementation, the confidence interval fluctuates around zero. This implies that there is no significant systematic difference in the number of patent authorizations between the treatment and control groups. Thus, we can proceed with the parallel trends hypothesis test.

**4.3.4 Placebo test.** Placebo testing serves as a robustness test to further validate the results by manipulating the timing of policy shocks [53]. To ensure that the growth of green innovation originates from the impact of LCPC policy rather than other unobservable factors, we conducted placebo testing. The methodology involved expanding the original dataset by a factor of 128 and randomly sampling from this expanded dataset without replacement each year, ensuring that the number of enterprises adopting the policy in the sampled data matched the corresponding year in the original sample. Subsequently, a DID regression was performed on the randomly sampled data to derive regression coefficients. This study employed extended data consisting of 100 samples and conducted placebo tests for the two dependent variables:

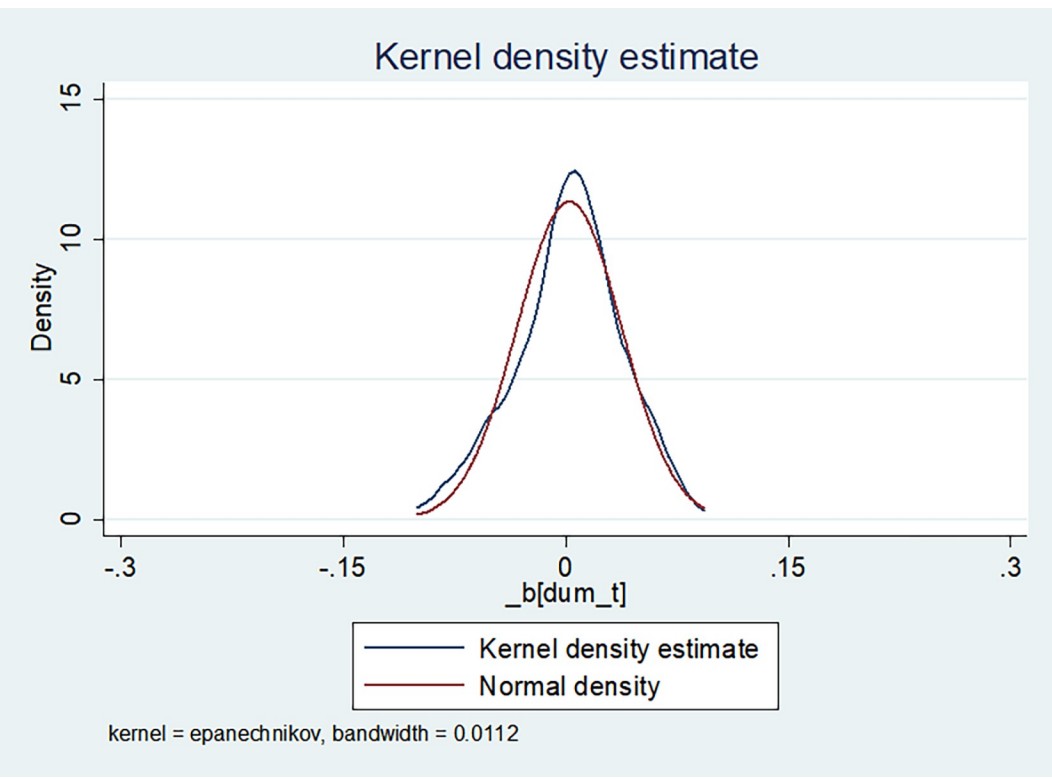

**Fig 2. Placebo test of the total number of green patent licenses.**

total green patent authorizations and green invention patent authorizations, as illustrated in Figs 2 and 3. Figs 2 and 3 show that the coefficient and density distribution plots. It is noticeable that the coefficients plot closely approximates the normal distribution curve, with the majority of coefficient values falling to the left of the true coefficient value (1.762 and 1.149). This signifies a significant enhancement in the green innovation capacity of enterprises resulting from the LCPC policy.

## 5 Heterogeneity and mechanism analysis

### 5.1 Heterogeneity test

In the preceding sections, we examined the overall impact of LCPC policy on the relationship between financial development and green innovation in enterprises using the entire sample. However, there are substantial variations in factors such as enterprise ownership structure, the presence of managerial myopia, industry affiliation, and geographic location. To gain a deeper understanding of how financial development influences green innovation in enterprises located within LCPC under different contextual conditions, we conducted additional analyses. Specifically, we conducted separate analyses considering variations in ownership structure, managerial myopia behavior, industry affiliation, and regional disparities.

**5.1.1 Heterogeneity of enterprise ownership.** We divided the sample into two sub-samples based on the ownership structure of the enterprises: state-owned and non-state-owned. We analyzed the impact of financial development on green innovation under the LCPC policy for these two types of enterprises, and the results are presented in Table 10. Columns (1) and (3) represent the results for state-owned enterprises, while columns (2) and (4) represent the

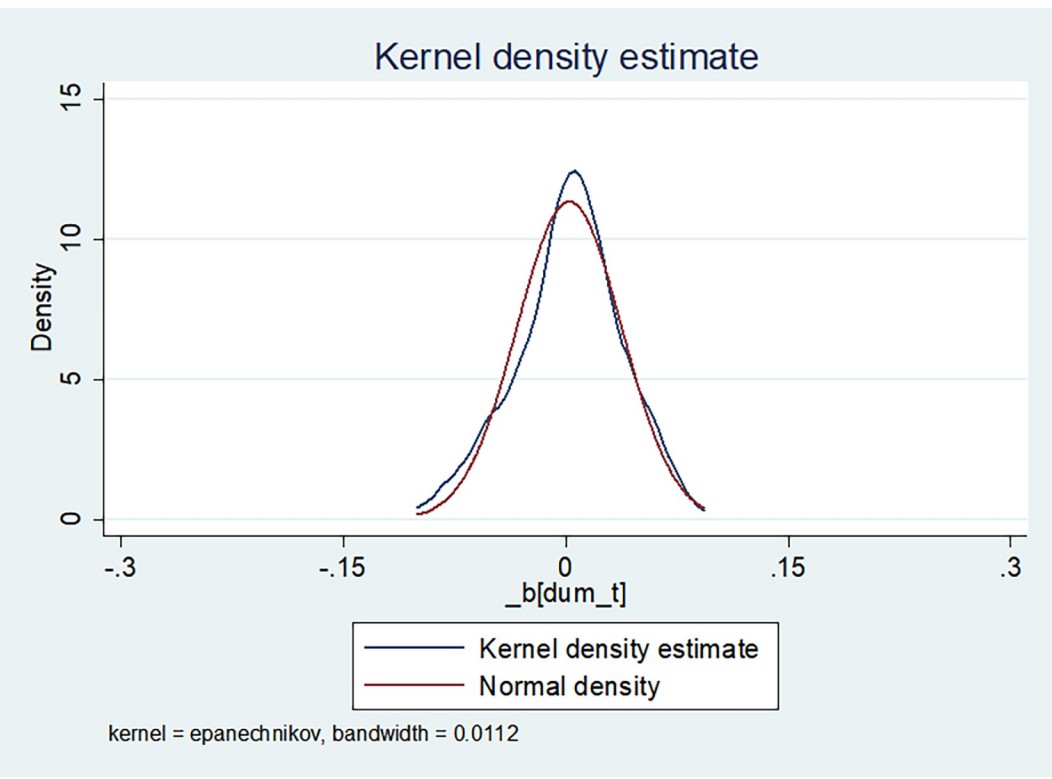

**Fig 3. Placebo test of the authorized amount of green invention patents.**

results for non-state-owned enterprises. The results indicate a significant effect of financial development on promoting green innovation in state-owned enterprises under the impact of the LCPC policy, whereas the effect on non-state-owned enterprises is less pronounced. This suggests that state-owned enterprises are more capable of implementing policy measures and

**Table 10. Group inspection of enterprise ownership.**

| | Patents | | Inventions | |
|---|---|---|---|---|
| | state-owned | Non-state | state-owned | Non-state |
| | (1) | (2) | (3) | (4) |
| Interactive item | -1.930** | -0.210 | -1.532* | -0.186 |
| | (-2.54) | (-1.11) | (-1.70) | (-1.31) |
| control variable | yes | yes | yes | yes |
| Fixed year effect | yes | yes | yes | yes |
| Individual fixed effect | yes | yes | yes | yes |
| Provincial fixed effect | yes | yes | yes | yes |
| _ cons | -0.389 | -0.293 | -0.389 | -0.293 |
| | (-0.10) | (-0.33) | (-0.10) | (-0.33) |
| N | 3969 | 6313 | 3969 | 6313 |
| $R^2$_a | 0.069 | 0.017 | 0.069 | 0.017 |

Note: The t value of the regression coefficient is shown in brackets, and
*, **, and ***represent the significance level of 10%, 5%, and 1% respectively.

are more inclined to engage in green technological innovation when financial resources are relatively easily accessible.

This difference in effects can be attributed to various factors. Firstly, state-owned enterprises play a crucial role in a country's economy and have access to more resources and government support. Therefore, they are more likely to obtain financial support and allocate resources toward green innovation. State-owned enterprises may have easier access to low-interest loans, government subsidies, or other financial incentives that support their green technological innovation efforts.

Furthermore, state-owned enterprises often hold dominant market positions and exhibit monopolistic tendencies within their industries, which results in relatively lower competitive pressures. This reduced competition may decrease the motivation for non-state-owned enterprises to pursue green innovation, thereby diminishing their willingness to seek financial support and invest in technological advancements. Moreover, non-state-owned enterprises may encounter challenges related to accessing financial resources and facing information asymmetry, which can restrict their ability to obtain financial support and necessary resources. These barriers and information gaps can impede non-state-owned enterprises from fully capitalizing on the opportunities provided by financial development to drive green innovation.

Additionally, cultural and managerial system disparities can also influence the green innovation efforts of state-owned and non-state-owned enterprises. State-owned enterprises often possess more stable organizational structures and centralized decision-making mechanisms, enabling them to make prompt and efficient decisions when implementing green innovation initiatives. Conversely, non-state-owned enterprises may face more obstacles in decision-making processes and cultural aspects, leading to slower progress in green innovation. Furthermore, state-owned enterprises often shoulder greater social responsibilities, including environmental protection and sustainable development, within their operational domains. With government oversight and guidance, it is easier for these enterprises to promote green innovation under the LCPC policy. In contrast, non-state-owned enterprises may experience less pressure to fulfill social responsibilities, resulting in reduced enthusiasm and investment in green innovation.

To summarize, there are disparities in the impact of financial development on green innovation between state-owned and non-state-owned enterprises. To foster green innovation in non-state-owned enterprises, the government, and financial institutions can implement relevant policies and measures. These may include increasing financial support, reducing barriers to entry, and enhancing industry collaboration and knowledge sharing. Through collective efforts, sustainable development can be propelled forward.

**5.1.2 Heterogeneity of management short-sightedness.** Based on the relationship between the number of patent inventions by companies and the presence of short-sighted behavior in their management, we divided the sample into two sub-samples: one comprising a company exhibiting short-sighted behavior and another comprising a company without such behavior. Furthermore, we analyzed whether the impact of financial development on green innovation differed for different management groups under the LCPC policy. The results are presented in Table 11, where columns (1) and (3) represent the results for companies with short-sighted behavior, and columns (2) and (4) represent the results for companies without short-sighted behavior. The findings indicate that under the influence of LCPC policy, financial development significantly influences the promotion of green innovation in companies with short-sighted behavior, but its effect on companies without short-sighted behavior is less pronounced. This suggests that companies with short-sighted behavior are more likely to implement policy execution and engage in innovative green technologies when funding is relatively accessible.

**Table 11. Management short-sightedness group test.**

| | Patents | | Inventions | |
|---|---|---|---|---|
| | Have short-sighted behavior | No short-sightedness | Have short-sighted behavior | No short-sightedness |
| | (1) | (2) | (3) | (4) |
| Interaction items | -1.017* | -1.404 | -0.907** | -0.546 |
| | (-1.78) | (-1.58) | (-1.97) | (-0.83) |
| Control variables | yes | yes | yes | yes |
| Year fixed effects | yes | yes | yes | yes |
| Individual fixed effects | yes | yes | yes | yes |
| Provincial fixed effects | yes | yes | yes | yes |
| _cons | -0.089 | 1.368 | -0.089 | 1.368 |
| | (-0.05) | (0.43) | (-0.05) | (0.43) |
| N | 8841 | 1441 | 8841 | 1441 |
| $R^2$ _a | -0.155 | -1.074 | -0.153 | -1.108 |

Note: The t value of the regression coefficient is shown in brackets, and

*, **, and ***represent the significance level of 10%, 5%, and 1% respectively.

Regarding the reasons for these results, we can analyze the following points:

Firstly, management with short-sighted behavior may prioritize short-term gains and be hesitant to invest in long-term green innovation. Financial development support can provide the necessary funding and resources, facilitating these companies in taking action toward green innovation and increasing motivation for low-carbon emission reduction.

Secondly, management with short-sighted behavior may lack relevant information and expertise in the field of green innovation, leading to a more noticeable response to the support of financial development. In contrast, management without short-sighted behavior may possess greater awareness and capability in green innovation, already taking proactive measures, which results in a lesser impact from financial development support. Additionally, management with short-sighted behavior may exhibit a higher aversion to risk and a lower willingness to take risks in green innovation. The support of financial development can mitigate the risks associated with green innovation, thus significantly influencing companies with short-sighted behavior.

In conclusion, within the context of LCPC policy, financial development significantly promotes green innovation in companies with short-sighted behavior, while its effect is relatively weaker for companies without such behavior. This finding holds significant importance in understanding the impact of internal management behavior on green innovation and in formulating targeted policy recommendations, thereby contributing to the advancement of green technological innovation and sustainable development.

**5.1.3 Heterogeneity of industry categories.** Generally, the number of patent inventions in non-high-tech industries should exceed that of high-tech industries. To investigate whether the influence of financial development on green innovation varies across industry categories, we categorized companies into high-tech and non-high-tech industries based on the definitions provided in the "Classification of High-tech Industries (Manufacturing) (2017)" and "Classification of High-tech Industries (Services) (2018)" issued by the National Bureau of Statistics. Companies with two-digit industry codes 26, 27, 30, 32, 33, 34, 35, 36, 37, 38, 39, 40, 46, 63, 64, 65, 73, 74, and 75 were classified as high-tech industries, while the remaining codes were assigned to non-high-tech industries.

As demonstrated in Table 12, financial development exerts a substantial influence on green innovation in non-high-tech industries. With the implementation of the LCPC policy, both

**Table 12. Group inspection by industry category.**

| | Patents | | Inventions | |
|---|---|---|---|---|
| | High-tech industry | Non-high technology industry | High-tech industry | Non-high technology industry |
| | (1) | (2) | (3) | (4) |
| Interactive item | -0.317 | -1.277* | -0.307 | -1.059* |
| | (-0.64) | (-1.69) | (-1.00) | (-1.65) |
| control variable | yes | yes | yes | yes |
| Fixed year effect | yes | yes | yes | yes |
| Individual fixed effect | yes | yes | yes | yes |
| Provincial fixed effect | yes | yes | yes | yes |
| _ cons | -0.089 | 1.368 | -0.089 | 1.368 |
| | (-0.05) | (0.43) | (-0.05) | (0.43) |
| N | 4172 | 3705 | 4172 | 3705 |
| $R^2$_a | -0.204 | -0.136 | -0.201 | -0.097 |

Note: The t value of the regression coefficient is shown in brackets, and

*, **, and ***represent the significance level of 10%, 5%, and 1% respectively.

the total number of green patents and the count of green invention patents in non-high-tech industries experienced significant growth, whereas the effect on high-tech industries was less pronounced. This indicates that financial development plays a crucial role in driving green innovation in non-high-tech industries during policy shocks. Consequently, non-high-tech industries increase their investment in technological innovation and emphasize the enhancement of green technologies. Several factors can account for these findings.

Firstly, non-high-tech industries generally possess limited resources for technological research and development. Financial development can offer funding and support to these industries, enabling them to strengthen their efforts in green innovation. Non-high-tech industries require relatively lower technological inputs compared to high-tech industries, making the impact of financial development more noticeable. Furthermore, non-high-tech industries often face less intense market competition pressures. The financial development initiatives under the LCPC policy create more opportunities and incentives for these industries, motivating them to actively engage in green innovation and enhance their competitiveness in the market.

Secondly, non-high-tech industries may have a greater demand for improvement and transformation. In contrast to high-tech industries that already operate at the forefront of technological innovation, non-high-tech industries may lag in the application and research of green technologies. In such cases, the impetus from financial development can provide stronger incentives and support for non-high-tech industries, propelling them to strengthen their efforts in green innovation.

In conclusion, under the LCPC policy, financial development significantly promotes green innovation in non-high-tech industries. When implementing policies and measures, the government and relevant stakeholders must reinforce initiatives such as financial support, technical training, and information sharing. These efforts will further advance green innovation in non-high-tech industries and facilitate the achievement of sustainable development.

**5.1.4 Heterogeneity of geographical regions.** The geographic region in which an enterprise operates directly influences its level of green innovation. In addition to the East-West divide, a North-South disparity has emerged as a recent regional development trend. Generally, companies in the South are expected to have higher levels of green innovation compared

to those in the North. To investigate whether the impact of financial development on green innovation varies across geographic regions, this study divided all enterprise samples into two groups: South and North, and conducted group regression analysis as presented in Table 13. Columns (1) and (3) represent the results for the South, while columns (2) and (4) represent the results for the North. The regression results indicate that financial development promotes green technological innovation in companies operating in the South following the implementation of the LCPC policy. However, no significant promotion effect is observed in the North. This suggests differences in green innovation between companies in the North and South of China, with companies in the South being more inclined to engage in technological innovation under the impetus of financial development. This is also one of the reasons for the widening gap in economic growth between the North and South of China.

The reasons for this heterogeneity are as follows: Firstly, the financial system in the South is more developed compared to the North. The South has a greater number and variety of financial institutions, resulting in a relatively more active financial market. This development in the financial system provides more financing channels and financial tools for companies in the South, making it easier for them to access financial support, including innovation investment funds, venture capital, green bonds, and others. The abundance and accessibility of these financial resources offer more opportunities and motivation for companies in the South to engage in green technological innovation.

Secondly, the South generally possesses a more comprehensive technological innovation ecosystem, including research institutions, universities, and innovation incubators, among others. These institutions and platforms provide better research and development resources for companies in the South, such as talent, technology consulting, and research collaborations. Moreover, the density of companies in the South has led to the formation of closely-knit industrial chains and innovation clusters, facilitating knowledge exchange and the incubation of innovation. The existence of such technological innovation ecosystems provides a more favorable environment for companies in the South to engage in green technological innovation.

Furthermore, the South experiences faster economic development and has a larger market size, resulting in a more urgent demand for green technological innovation. Companies in the South face market demands and competitive pressures, which incline them to engage in

**Table 13. Sample inspection by geographical region.**

|  | Patents | | Inventions | |
|---|---|---|---|---|
|  | south | north | south | north |
|  | (1) | (2) | (3) | (4) |
| Interactive item | -0.812* | -1.535 | -0.332* | -1.369 |
|  | (-1.84) | (-1.46) | (-1.94) | (-1.38) |
| control variable | yes | yes | yes | yes |
| Fixed year effect | yes | yes | yes | yes |
| Individual fixed effect | yes | yes | yes | yes |
| Provincial fixed effect | yes | yes | yes | yes |
| _ cons | -1.034 | -0.264 | -1.034 | -0.264 |
|  | (-0.92) | (-0.14) | (-0.92) | (-0.14) |
| N | 4555 | 3322 | 4555 | 3322 |
| $R^2$_a | 0.017 | 0.065 | 0.030 | 0.063 |

Note: The t value of the regression coefficient is shown in brackets, and

*, **, and ***represent the significance level of 10%, 5%, and 1% respectively.

technological innovation under the impetus of financial development to meet market needs, enhance competitiveness, and achieve sustainable development.

Lastly, the governments in the South generally place a greater emphasis on environmental protection and sustainable development and have implemented a series of policy measures to support green technological innovation. These policy measures include fiscal support, tax incentives, subsidies, and reward mechanisms, among others, which encourage companies to engage in green technological innovation. Companies in the South are more likely to benefit from these policy supports, further strengthening their willingness to engage in technological innovation under the impetus of financial development.

In conclusion, the differences between the North and South of China in terms of financial systems, technological ecosystems, and market demands result in a higher inclination of companies in the South to engage in technological innovation under the impetus of financial development to meet market needs, enhance competitiveness, and achieve sustainable development goals. In contrast, companies in the North may be constrained by various factors, leading to a relatively weaker inclination toward green technological innovation.

## 5.2 Mechanism analysis

From column (1) of Table 14, it is evident that there is a positive correlation between the concentration of bank branches and the interaction between LCPC policy and financial constraints. This suggests that the implementation of LCPC policy, coupled with financial development, can mitigate financing constraints for companies and facilitate the promotion of green innovation.

LCPC policy provides crucial policy support and guidance for green innovation. Governments can formulate relevant policies such as tax incentives, fiscal subsidies, and green innovation funds to incentivize financial institutions to provide financial support to green innovation companies. Additionally, financial institutions can introduce innovative financial products, including green bonds, sustainable development loans, and environmental protection insurance, specifically tailored to cater to the financing needs of green innovation companies. These financial products effectively reduce financing costs and risks, thereby enhancing the financing capacity of green innovation companies.

Also, financial institutions can establish robust green risk management mechanisms to evaluate and monitor green innovation projects. Through the adoption of scientific evaluation

**Table 14. Mechanism analysis.**

|  | (1) | (2) |
|---|---|---|
|  | FC | R & D |
| Interaction items | 0.001*** | 0.340** |
|  | (3.21) | (0.172) |
| Control variables | yes | yes |
| Year fixed effects | yes | yes |
| Individual fixed effects | yes | yes |
| Provincial fixed effects | yes | yes |
| _cons | -3.6e+11*** | -8.6e+07 |
|  | (-7.85e+13) | (-0.74) |
| N | 7877 | 7827 |
| $R^2$ _a | 1.000 | 0.074 |

Note: The t value of the regression coefficient is shown in brackets, and

*, **, and ***represent the significance level of 10%, 5%, and 1% respectively.

methods and risk analysis, financial institutions can alleviate uncertainty and risk perception associated with green innovation projects, consequently mitigating financing constraints. Moreover, collaborative platforms can be established between financial institutions and green innovation companies in LCPC, facilitating information sharing and exchange. By strengthening collaborative relationships, financial institutions gain a deeper understanding of the needs and challenges faced by green innovation companies, enabling them to provide customized financial services that further alleviate financing constraints.

Furthermore, financial institutions can actively fulfill their social responsibilities in promoting the development of green innovation. For instance, they can enhance the promotion and training of green finance to increase public awareness and understanding of green innovation. This initiative serves to enhance the visibility and reputation of green innovation companies, ultimately leading to a reduction in financing constraints.

In column (2) of Table 14, we have also confirmed the positive impact of financial development on increasing the level of research and development (R&D) investment in companies. An increase in the proportion of R&D investment signifies a higher level of R&D investment, which in turn drives companies towards adopting green production methods, fostering the advancement of green development.

Firstly, financial development offers companies a wider range of financing channels and financial tools, facilitating their access to funding support. These include innovative investment funds, venture capital, and green bonds, among others. By securing increased financial support, companies can allocate more resources towards green technology R&D, thus accelerating the pace of green innovation.

Secondly, financial development also fosters the establishment of technological innovation ecosystems. With improved financial development, research institutions, universities, and innovation incubators, among other technological innovation entities, gain enhanced access to resources and development opportunities. These entities provide companies with R&D and innovation resources, such as skilled personnel, technology consulting, and research collaborations. Collaborating with these institutions allows companies to leverage shared resources and knowledge, driving R&D and innovation in green technologies.

Thirdly, financial development can incentivize companies to augment their investment in green technology innovation by introducing R&D investment incentives. For instance, governments can provide R&D subsidies, rewards, and tax incentives, among other policy measures, to mitigate R&D costs and risks for companies. This encourages companies to increase their investment in green technology innovation.

The analysis process outlined above is summarized (see Fig 4). This mechanism enables LCPC to alleviate financing constraints and boost R&D investment levels, thereby affording enterprises more opportunities and incentives to foster the robust development of green innovation activities. The collaboration efforts of the governments, financial institutions, and companies establish a conducive policy environment and financial ecosystem, offering substantial financial backing for green innovation and propelling the sustainable advancement of low-carbon economies.

# 6 Conclusions and recommendations

## 6.1 Conclusions and contributions

Based on data from listed companies and prefecture-level cities spanning 2010 to 2018, this study integrates economic, management, and organizational psychology theories to delve into the profound impact of financial development on green innovation within enterprises, operating under the LCPC policy. Additionally, it examines the moderating roles of financing

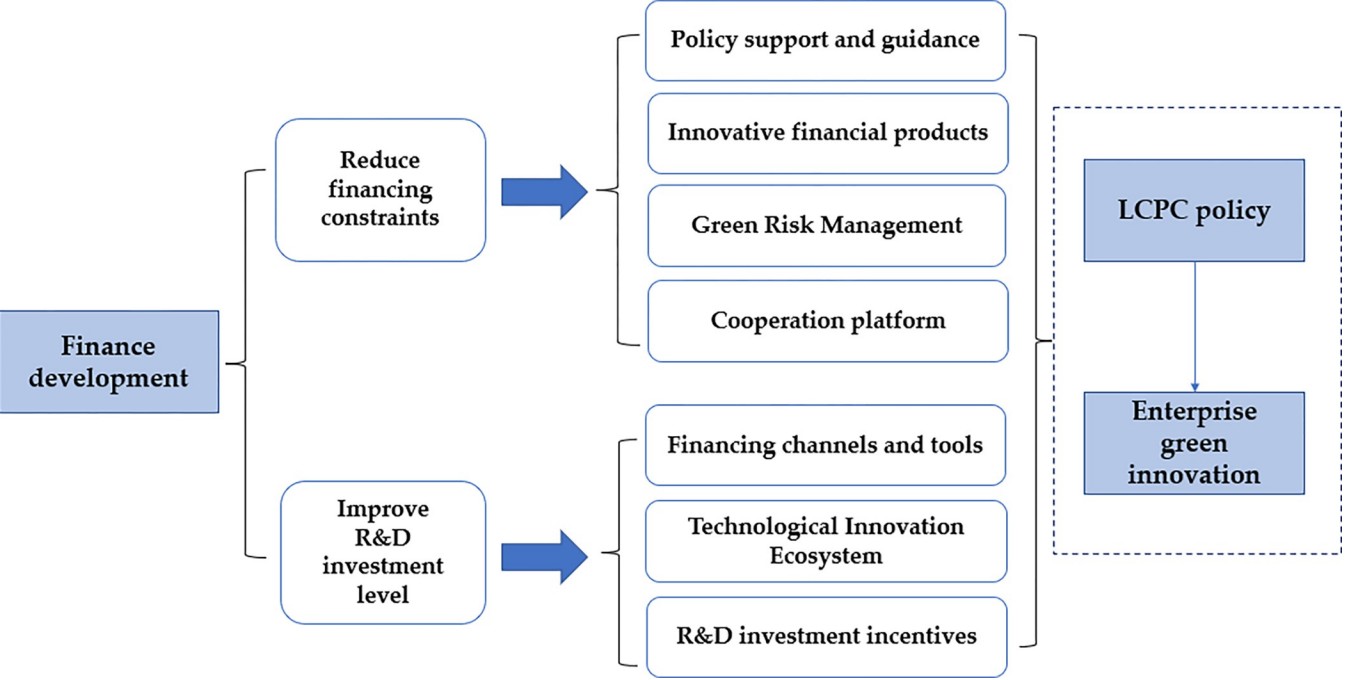

**Fig 4. The mechanisms of financial development's impact on corporate green innovation.**

constraints and R&D investment. The findings indicate that (1) LCPC policy positively influences enterprise green innovation, a result confirmed through rigorous robustness tests, with financial development further amplifying this positive effect;(2) the impact of financial development on enterprise green innovation is more pronounced when firms exhibit heterogeneous ownership, managerial short-sightedness, and specific industry or regional characteristics; and (3) detailed analyses reveal that augmenting R&D investment and alleviating financing constraints effectively moderate the relationship between financial development and corporate green innovation. Specifically, the negative impact of financial development on corporate green innovation is heightened under the LCPC policy.

Our primary contributions can be delineated in the following facts.:

Firstly, adopting perspectives from organizational psychology and behavior, it scrutinizes the influence of the level of financial development on the green innovation of low-carbon pilot enterprises, thereby pushing the theoretical boundaries of environmental policy research. While prior literature has acknowledged the crucial relationship between environmental policy and corporate green innovation, most studies have been confined to accentuating the "favorable or unfavorable" connection between the two, neglecting adequate attention to the conditions determining favorability, unfavorability, and the temporal aspects of these potential effects. The economic impact of environmental policies currently sparks controversy. One viewpoint suggests that stringent environmental regulations might stimulate firms' innovative behavior. As per the "Porter hypothesis" [10], strict environmental regulations may incentivize firms towards innovation more than the cost of compliance, prompting a heightened focus on innovation. However, the static analytical framework of perfectly competitive markets in neoclassical economics posits that environmental policies are not inherently linked to firm innovation [11]. Existing research leans heavily on theoretical extrapolations and qualitative studies, lacking empirical evidence support. Consequently, the discourse on the relationship between environmental regulations and corporate green innovation remains constrained by a

dearth of research contexts. This paper empirically examines the impact of LCPC policies on corporate green innovation. In doing so, it not only broadens the spectrum of influencing factors for green innovation but also contributes incrementally to advancing research in the realm of environmental regulations within the context of corporate green innovation.

Secondly, this study delves into the boundary conditions influencing the impact of environmental regulation on corporate green innovation through a thorough investigation of financial development. Many previous studies have adopted a simplistic dichotomous approach, solely discussing the direct impact of environmental regulation on corporate green innovation. However, much of the research on LCPC policies and corporate green innovation has been confined to a singular perspective, overlooking the exploration of boundary conditions between the two. This paper unveils the boundary conditions of the impact of LCPC policy on corporate green innovation by introducing financial development as a typical and crucial factor. It adopts the perspective of the technological innovation mechanism emphasized by Porter's hypothesis. This approach aids in fostering the integration of diverse theories in environmental regulation research, introduces new research perspectives on related issues in this field, and furnishes substantial empirical support for future implementations of financial development and low-carbon policies.

## 6.2 Recommendations

Following the aforementioned conclusions and discussions, to further advance the development of a low-carbon economy and achieve a comprehensive green transformation of the economy and society, we propose the following countermeasures and recommendations:

Firstly, the government can implement more concrete incentives, including substantial tax incentives and fiscal subsidies, to actively encourage financial institutions to provide financing support for green innovation enterprises. These tangible incentives will significantly reduce the financing costs for enterprises, thereby enhancing their motivation to participate in green innovation. At the same time, financial institutions are urged to develop specialized financial products for green innovative enterprises, such as green bonds, sustainable development loans, and environmental insurance. These innovative financial products offer more flexible and responsive financing options tailored to meet the needs of green innovative enterprises.

Secondly, financial institutions should establish a comprehensive green risk management mechanism to systematically mitigate uncertainty and risk perception associated with green innovation projects. This can be achieved by scientific assessment methods and thorough risk analysis, ultimately alleviating financing constraints. Concurrently, financial institutions should establish a collaboration platform with green innovation enterprises in LCPC to facilitate information sharing and exchange. Actively engaging in this collaboration enables financial institutions to gain deeper insights into the needs and challenges of green innovation enterprises, allowing them to provide personalized financial services and effectively reduce financing constraints. Thirdly, financial institutions should intensify efforts in publicizing and training green finance to enhance public awareness and understanding of green innovation. Improved understanding among investors and lending institutions can enhance support and financing opportunities for green innovative enterprises. Additionally, close collaboration between the government and financial institutions is crucial for establishing a specialized green innovation investment fund dedicated to supporting enterprises in LCPC. This fund would not only offer venture capital but also provide capital support for green innovation projects, facilitating their development and implementation.

In summary, these recommendations are designed to bolster financial support for green innovation in enterprises within LCPC. They aim to strengthen financing channels, alleviate

financing constraints, and facilitate the widespread implementation and promotion of green innovation. Collaboration among governments, financial institutions, and enterprises is essential to construct a robust financial ecosystem, positively contributing to the sustainable development of the low-carbon economy.

## 6.3 Limitations

This paper acknowledges certain research limitations and identifies areas for potential expansion. Initially, the study relies on data spanning from 2010 to 2018. Considering the evolving socio-economic landscape, especially with the recent strides in the global green transition, future investigations may benefit from utilizing more recent data to capture emerging trends comprehensively. While the research examines the impact of financial development on low-carbon pilot city policies and corporate green innovation, there is room for refinement. Subsequent studies could delve into the nuanced differences in corporate green innovation support from distinct financial institutions, such as commercial banks and investment funds. Furthermore, the recommendation for governments to offer substantial tax incentives and financial subsidies to encourage financial institutions to support green innovative enterprises involves considerations of government fiscal policies and budget constraints. Future research could explore the feasibility of policies in this realm and the associated implementation challenges in greater detail. In the context of recommendations on green finance publicity and training, future research could quantitatively assess the actual impact of these campaigns on public awareness and understanding. This quantitative analysis aims to provide a more comprehensive understanding of the role of such initiatives in promoting green innovation.

In conclusion, these acknowledged research limitations open avenues for future exploration, offering opportunities to delve more deeply into the relationship between enterprise green innovation and financial development. This exploration is crucial for fostering a more sustainable green economy.

## Author Contributions

**Conceptualization:** Jianxiao Du.

**Data curation:** Jianxiao Du, Xiaoyu Cui.

**Formal analysis:** Yajie Han.

**Funding acquisition:** Yajie Han.

**Methodology:** Jianxiao Du.

**Supervision:** Xiaoyu Cui.

**Writing – original draft:** Jianxiao Du, Yajie Han.

**Writing – review & editing:** Yajie Han, Xiaoyu Cui.

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
