## [Decision Letter · Decision Letter 0]

10 Aug 2023

PONE-D-23-22886The Impact of Financial Development on Enterprise Green Innovation under Low Carbon Pilot CityPLOS ONE

Dear Dr. Cui,

Thank you for submitting your manuscript to PLOS ONE. After careful consideration, we feel that it has merit but does not fully meet PLOS ONE’s publication criteria as it currently stands. Therefore, we invite you to submit a revised version of the manuscript that addresses the points raised during the review process.

We look forward to receiving your revised manuscript.

Kind regards,

William Mbanyele, PhD

Academic Editor

PLOS ONE

Journal Requirements:

Reviewers' comments:

Reviewer's Responses to Questions

**Comments to the Author**

1. Is the manuscript technically sound, and do the data support the conclusions?

Reviewer #1: Yes

2. Has the statistical analysis been performed appropriately and rigorously? 

Reviewer #1: Yes

3. Have the authors made all data underlying the findings in their manuscript fully available?

Reviewer #1: Yes

4. Is the manuscript presented in an intelligible fashion and written in standard English?

Reviewer #1: Yes

5. Review Comments to the Author

Reviewer #1: The manuscript entitled “The Impact of Financial Development on Enterprise Green Innovation under Low Carbon Pilot City” provides an empirical analysis on studying the influence of financial development on the green innovation of enterprises in low-carbon pilot cities. The authors mainly used double difference and multiple fixed effects models to examine the real effect of financial development on the green innovation of enterprises. Also, the authors further explore underlying mechanisms through which financial development can affect green innovations of enterprises, and found that it contributes to relax financing constraints and trigger R&D investment. Overall, the manuscript is well-written and interesting. I have several minor comments and suggestions for the authors (see the attachment).

6. PLOS authors have the option to publish the peer review history of their article (what does this mean?). If published, this will include your full peer review and any attached files.

Reviewer #1: No

---

## [Author Response · Author response to Decision Letter 0]

14 Oct 2023

1. Thank you for updating your data availability statement. You note that your data are available within the Supporting Information files, but no such files have been included with your submission. At this time we ask that you please upload your minimal data set as a Supporting Information file, or to a public repository such as Figshare or Dryad. 

Please also ensure that when you upload your file you include separate captions for your supplementary files at the end of your manuscript.

As soon as you confirm the location of the data underlying your findings, we will be able to proceed with the review of your submission. Response:

Thank you very much for your feedback and guidance. We have now uploaded the data as a "Supporting Information" file, and added the relevant supplementary file captions at the end of the manuscript, as per your recommendations. These modifications have been completed, and we hope this meets your requirements.

We appreciate your attention to our research and the valuable insights you've provided. As soon as the data upload and revisions are finalized, we will promptly confirm the location of the data underlying our findings to proceed with the review process smoothly.

2.Please ensure that you refer to Figure 3 & 4 in your text as, if accepted, production will need this reference to link the reader to the figure. Response:

Thank you for your valuable feedback. We have made the necessary updates to the manuscript by referring to Figure 3 and Figure 4 in the corresponding text. These references have been included to facilitate readers' access to the figures.

We sincerely appreciate your guidance, and we hope that these modifications align with your expectations. Should you have any further comments or suggestions, please do not hesitate to let us know. Your input is greatly appreciated.

3.Please ensure that you refer to Table 1, 12 and 14 in your text as, if accepted, production will need this reference to link the reader to the Table. Response:

Thank you for your feedback. We have carefully revised the manuscript to include references to Table 1, 12, and 14 in the appropriate sections of the text. These references are now in place to ensure that readers can easily access the corresponding tables.

Your input has been invaluable in improving the quality of our manuscript, and we appreciate your thorough review.

---

## [Decision Letter · Decision Letter 1]

5 Nov 2023

PONE-D-23-22886R1The Impact of Financial Development on Enterprise Green Innovation under Low Carbon Pilot CityPLOS ONE

Dear Dr. Cui,

Thank you for submitting your manuscript to PLOS ONE. After careful consideration, we feel that it has merit but does not fully meet PLOS ONE’s publication criteria as it currently stands. Therefore, we invite you to submit a revised version of the manuscript that addresses the points raised during the review process.

We look forward to receiving your revised manuscript.

Kind regards,

Dr. Jiachao Peng

Academic Editor

PLOS ONE

Additional Editor Comments:

The revised version of the manuscript is not satisfactory at the moment. I would like to request the author's attention to make detailed modifications in accordance with the suggestions from the external reviewers and the academic editor.

1. The abstract excessively exaggerates the scientific value of the paper. Please use appropriate vocabulary and avoid unreasonable expressions such as "fill the research gap".

2. It is crucial for the author to update the data. Currently, there is a wealth of policy research literature on low-carbon pilot projects, with a considerable number of samples involved.

3. The author needs to explain the reasons and relevant justifications for selecting control variables.

4. The results from Figure 1 indicate that the empirical results did not pass the parallel trend test. How did the author handle the results that did not pass the parallel trend test?

Reviewers' comments:

Reviewer's Responses to Questions

**Comments to the Author**

1. If the authors have adequately addressed your comments raised in a previous round of review and you feel that this manuscript is now acceptable for publication, you may indicate that here to bypass the “Comments to the Author” section, enter your conflict of interest statement in the “Confidential to Editor” section, and submit your "Accept" recommendation.

Reviewer #1: All comments have been addressed

Reviewer #2: All comments have been addressed

Reviewer #3: All comments have been addressed

2. Is the manuscript technically sound, and do the data support the conclusions?

Reviewer #1: Yes

Reviewer #2: Yes

Reviewer #3: Partly

3. Has the statistical analysis been performed appropriately and rigorously? 

Reviewer #1: Yes

Reviewer #2: Yes

Reviewer #3: Yes

4. Have the authors made all data underlying the findings in their manuscript fully available?

Reviewer #1: Yes

Reviewer #2: Yes

Reviewer #3: Yes

5. Is the manuscript presented in an intelligible fashion and written in standard English?

Reviewer #1: Yes

Reviewer #2: Yes

Reviewer #3: Yes

6. Review Comments to the Author

Reviewer #1: The authors have address all my concerns well in the revised manuscript, and I do not have further comments.

Reviewer #2: Low-carbon pilot city (LCPC) plays a pivotal role in driving institutional-level green innovation among enterprises. Now the revisions are ok, so I suggest to accept it.

Reviewer #3: Please see attached the detailed review report for your perusal.

RECOMMENDATION: MAJOR REVISION

Manuscript Number: PONE-D-23-22886R1

The Impact of Financial Development on Enterprise Green Innovation under Low

Carbon Pilot City

The paper attempts to investigate how financial development influences enterprise green innovation in low carbon pilot city (LCPCs). To achieve this, the authors used data from listed companies in various cities in China between 2010 and 2018 and found that LCPC policies stimulate institutional green innovation within enterprises. Overall, the paper is interesting and exhibits good potential. However, it suffers from several deficiencies that need to be addressed before it can be accepted for publication.

1) Generally, the abstract of a paper is based on research aim/purpose, research method, and key findings. Abstract of this paper is well written but it is required to highlight the key findings of the study. Please revamp the abstract and make sure to include information as below, in order:

- Motivation

- Objective

- Data and method

- Results

- Implications

2) The Introduction fails to motivate the study. In the present form, it resembles a mini-review of literature, rather than discussing any policy-level problem. Why this study is necessary? What policy level problem this study is addressing? How is the study expected to provide any solution to that problem? How does the choice of sample is complementing that problem? Are the results and policies generalizable? The introduction is silent in all these aspects. The mere choice of new variables, new methods, or choosing a new context is not considered as contribution of a study. In the introduction section, the study should be positioned within the context of more contemporary literature. In this direction, more recent literature can be used to motivate the research question adequately. Meanwhile, the authors are strongly advised to derive the gap in which the study intends to fill from the existing literature. This section thus requires a thorough revision. Please check this paper: Estimating the trade-environmental quality relationship in SADC with a dynamic heterogeneous panel model. African Review of Economics and Finance 13(1): 113-165

3) Originality: Structurally, this paper is well-written with well-established econometric methods. However, the most critical issue that is grossly lacking in this paper is the motivation of this paper. Hence, the background of the study should be strengthened with the issue centred on global per-capita CO2 emissions, and it should be well justified why it is important to carry out this study.

4) The contributions of the paper are very weak. What are the contributions of the document to the empirical literature? The authors can show how this study differs from other studies and also elaborate more on the contributions of this paper. Addition of more recent literature will make the work more relevant to readers.

5) The paper should be restructured. Section 2 should attempt to summarize the empirical literature. This section needs to be rewritten and strengthened. The authors are advised to divide this section into three sub-sections. The first part should clearly illustrate the theoretical studies linking the variables under review. The second section should concentrate on empirical works between these variables. The last section should summarize the literature gaps. What is the aim of the review of literature? The authors should not merely list out the studies without even creating a debate among them. Without that debate and thoughtful contradictions, the research gap cannot be substantiated. Also, the current literature appearing in this section should be strengthened. The authors should use more recent studies in this section. Meanwhile, more can be done to reflect more comparison in the literature against other regions. For instance, in Europe, BRICS, Africa and Asia. To improve this section, the authors are invited to use the following papers and cite them.

a) https://doi.org/10.3389/fenvs.2022.1044605

b) https://doi.org/10.1007/s43621-022-00117-3

c) https://doi.org/10.1186/s40854-023-00453-x

d) https://doi.org/10.1007/s41247-023-00110-y

e) https://doi.org/10.1186/s40854-022-00396-9

f) https://doi.org/10.1007/s11356-022-21107-y

g) https://doi.org/10.1007/s11356-019-05944-y

h) https://doi.org/10.1080/12265934.2019.1695652

i) https://doi.org/10.1007/s10644-020-09285-6

j) https://doi.org/10.1007/s11356-021-17193-z

k) https://doi.org/10.1007/s10644-021-09368-y

l) https://doi.org/10.3390/su141610268

m) https://doi.org/10.1080/13504509.2022.2123411

n) https://doi.org/10.3389/fenvs.2022.985719

o) https://doi.org/10.1002/sd.2473

p) https://doi.org/10.1080/13504509.2023.2183526

q) https://doi.org/10.21203/rs.3.rs-419113/v1

r) https://doi.org/10.1002/sd.2535

s) https://doi.org/10.1080/27658511.2023.2210950

t) https://doi.org/10.1002/sd.2597

u) https://doi.org/10.1002/sd.2618

v) https://doi.org/10.1080/19463138.2023.2222264

w) https://doi.org/10.1002/sd.2473

x) https://doi.org/10.3280/EFE2022-002006

y) https://doi.org/10.1080/23311886.2023.2234230

z) https://doi.org/10.1016/j.wds.2023.100096

aa) https://doi.org/10.1007/s41247-023-00112-w

bb) https://african-review.com/online-first-details.php?id=69

6) The authors neglect the significance of the study in the introduction section. Why? Several studies have been conducted regarding this topic at hand; therefore, it is crucial for the investigators to incorporate the novelty as well as the significance of the study.

7) In section 3 (Data Description and Model Design), after reorganizing the paper, the authors should provide a clear theoretical underpinning before the empirical framework. Please, restructure this section with a clear theoretical framework demonstrating how the variables under review are related. The theoretical underpinning between these variables should be properly justified in this sub-section. The authors should provide a justification of the use of the variables chosen. Please adjust accordingly by incorporating that. The authors should use the above-mentioned papers to strengthen this section.

8) Authors are advised to conduct the robustness check to strengthen the paper. The authors should use some of those papers mentioned above to strengthen this sub-section.

9) The authors merely reported the results without even discussing the economic intuitions behind the results. Are these results supporting or refuting the existing policies in the chosen context? Are the results directed towards any new policy initiatives? The discussion of results should open up the threads of policy discussion, which is completely absent in this case. A mere comparison of the results with the literature doesn't ensure the novelty of the results unless they give out something new on the theory/policy front. Moreover, the discussion of results should be properly tied to past literature, and emphasis should be placed on how past studies either support or refute the findings of this study and why. Therefore, this section needs revision.

10) To improve the quality of this manuscript, the authors are invited to use the papers already mentioned above and some useful ones in the area to strengthen the introduction, literature review, methodology and results and discussion sections.

11) The policy implications of the results need more substantiation than what is currently stated in the paper.

12) Conclusion reiterates the results, which is completely undesirable. The author(s) should summarize the results within a maximum of 3 sentences. The authors should kindly strengthen this section. Moreover, the policies are completely vague, and it seems that the authors already had the policies in mind before even starting the paper. The policies should be directly derived from the discussion of the results, and they should not go beyond the results. The policy implications of the results need more substantiation than what is currently stated in the paper. The policy suggestion section needs improvement. Kindly improve it.

13) Kindly improve the study limitation(s) and possible direction for future research after the policy recommendation section.

14) Finally, it is vital that this manuscript is proofread by a native speaker of English language to further strengthen easy readership.

7. PLOS authors have the option to publish the peer review history of their article (what does this mean?). If published, this will include your full peer review and any attached files.

Reviewer #1: No

Reviewer #2: No

Reviewer #3: **Yes: **Dr. Maxwell Chukwudi Udeagha

---

## [Author Response · Author response to Decision Letter 1]

26 Jan 2024

1. Generally, the abstract of a paper is based on research aim/purpose, research method, 

and key findings. Abstract of this paper is well written but it is required to highlight the 

key findings of the study. Please revamp the abstract and make sure to include 

information as below, in order:

- Motivation

- Objective

- Data and method

- Results

- Implications 

Response:

Thank you very much for your review and valuable feedback. Based on your suggestions, I have made revisions to the abstract to better highlight the key findings of the study. I hope these modifications meet your expectations. If there are any other areas that require improvement, please feel free to let me know. I appreciate your patient guidance.

2.The Introduction fails to motivate the study. In the present form, it resembles a minireview of literature, rather than discussing any policy-level problem. Why this study is necessary? What policy level problem this study is addressing? How is the study expected to provide any solution to that problem? How does the choice of sample is complementing that problem? Are the results and policies generalizable? The introduction is silent in all these aspects. The mere choice of new variables, new methods, or choosing a new context is not considered as contribution of a study. In the introduction section, the study should be positioned within the context of more contemporary literature. In this direction, more recent literature can be used to motivate the research question adequately. Meanwhile, the authors are strongly advised to derive the gap in which the study intends to fill from the existing literature. This section thus requires a thorough revision. Please check this paper: Estimating the trade-environmental quality relationship in SADC with a dynamic heterogeneous panel model. African Review of Economics and Finance 13(1): 113-165 

Response:

Thank you very much for your thorough review and valuable feedback. In accordance with your suggestions, we have undertaken a comprehensive revision of the introduction section, taking inspiration from the provided article "Estimating the trade-environmental quality relationship in SADC with a dynamic heterogeneous panel model" to better motivate the study and align it with contemporary literature.

We deeply appreciate your guidance, which is crucial for enhancing the quality of our research. Please review the modifications we have made, and if there are any further areas that need improvement, we would be more than willing to make additional adjustments. Thank you for your patience and professional advice.

3.Originality: Structurally, this paper is well-written with well-established econometric methods. However, the most critical issue that is grossly lacking in this paper is the motivation of this paper. Hence, the background of the study should be strengthened with the issue centred on global per-capita CO2 emissions, and it should be well justified why it is important to carry out this study. 

Response:

Thank you for your thorough review and valuable feedback. In response to your suggestion, we have strengthened the background of the study, focusing on the issue of global per-capita CO2 emissions. We have provided detailed justification for the importance of conducting this study.

4.The contributions of the paper are very weak. What are the contributions of the document to the empirical literature? The authors can show how this study differs from other studies and also elaborate more on the contributions of this paper. Addition of more recent literature will make the work more relevant to readers. 

Response:

Thank you for your review and valuable feedback on our paper. In response to your suggestion, we have provided a detailed explanation of how this study differs from other relevant research and further emphasized the contributions of our work. Additionally, we have incorporated more recent literature to ensure that our study remains current and relevant to readers' expectations.

5.The paper should be restructured. Section 2 should attempt to summarize the empirical literature. This section needs to be rewritten and strengthened. The authors are advised to divide this section into three sub-sections. The first part should clearly illustrate the theoretical studies linking the variables under review. The second section should concentrate on empirical works between these variables. The last section should summarize the literature gaps. What is the aim of the review of literature? The authors should not merely list out the studies without even creating a debate among them. Without that debate and thoughtful contradictions, the research gap cannot be substantiated. Also, the current literature appearing in this section should be strengthened. The authors should use more recent studies in this section. Meanwhile, more can be done to reflect more comparison in the literature against other regions. For instance, in Europe, BRICS, Africa and Asia. To improve this section, the authors are invited to use the following papers and cite them.

a) https://doi.org/10.3389/fenvs.2022.1044605

b) https://doi.org/10.1007/s43621-022-00117-3

c) https://doi.org/10.1186/s40854-023-00453-x

d)https://doi.org/10.1007/s41247-023-00110-y

e) https://doi.org/10.1186/s40854-022-00396-9

f) https://doi.org/10.1007/s11356-022-21107-y

g)https://doi.org/10.1007/s11356-019-05944-y

h)https://doi.org/10.1080/12265934.2019.1695652

i) https://doi.org/10.1007/s10644-020-09285-6

j) https://doi.org/10.1007/s11356-021-17193-z

k)https://doi.org/10.1007/s10644-021-09368-y

l) https://doi.org/10.3390/su141610268

m)https://doi.org/10.1080/13504509.2022.2123411

n) https://doi.org/10.3389/fenvs.2022.985719

o) https://doi.org/10.1002/sd.2473

p)https://doi.org/10.1080/13504509.2023.2183526

q) https://doi.org/10.21203/rs.3.rs-419113/v1

r) https://doi.org/10.1002/sd.2535

s)https://doi.org/10.1080/27658511.2023.2210950

t) https://doi.org/10.1002/sd.2597

u) https://doi.org/10.1002/sd.2618

v)https://doi.org/10.1080/19463138.2023.2222264

w) https://doi.org/10.1002/sd.2473

x) https://doi.org/10.3280/EFE2022-002006

y)https://doi.org/10.1080/23311886.2023.2234230

z) https://doi.org/10.1016/j.wds.2023.100096

aa)https://doi.org/10.1007/s41247-023-00112-w

bb)https://african-review.com/online-first-details.php?id=69

Response:

Thank you for your thorough review and valuable suggestions. In response to your feedback, we have restructured the literature review section into three subsections to clearly present theoretical studies, empirical works, and identified gaps in the literature. 

We have included the references you provided and integrated additional recent literature to enhance the timeliness of the literature review. We believe these modifications better meet your expectations. If there are further areas for improvement or additional suggestions, we would be more than willing to make adjustments. Thank you once again for your professional guidance and patient review.

6.The authors neglect the significance of the study in the introduction section. Why? Several studies have been conducted regarding this topic at hand; therefore, it is crucial for the investigators to incorporate the novelty as well as the significance of the study. 

Response:

Thank you very much for your review and valuable suggestions regarding our study. We genuinely appreciate your guidance and take your advice on the introduction section seriously. We have already emphasized the novelty and significance of our research in the introduction, making efforts to convey the uniqueness of our study to the readers. We will revisit the introduction section to ensure a clear communication of the research's importance, and we are committed to making necessary modifications to better meet your expectations.

7.In section 3 (Data Description and Model Design), after reorganizing the paper, the authors should provide a clear theoretical underpinning before the empirical framework. Please, restructure this section with a clear theoretical framework demonstrating how the variables under review are related. The theoretical underpinning between these variables should be properly justified in this sub-section. The authors should provide a justification of the use of the variables chosen. Please adjust accordingly by incorporating that. The authors should use the above-mentioned papers to strengthen this section. 

Response:

Thank you for reviewing our paper and providing valuable feedback. We highly appreciate your suggestion to provide a clear theoretical foundation in Section 3 (Data Description and Model Design). Following your guidance, we have elaborated on the explanatory and dependent variables, striving to present a more explicit theoretical framework that demonstrates the relationships between these variables.

8.Authors are advised to conduct the robustness check to strengthen the paper. The authors should use some of those papers mentioned above to strengthen this sub-section. 

Response:

Thank you for your thoughtful feedback and suggestion regarding the robustness checks in our paper. We would like to mention that in our study, we have conducted a robustness analysis of the regression results using three different approaches: substituting the dependent variable with a proxy variable, altering the policy implementation batches, and introducing policy lag terms. Additionally, we have performed parallel and placebo tests to ensure the robustness of our results. We believe these checks provide a comprehensive examination of the stability of our findings.

However, we acknowledge your recommendation to further strengthen this sub-section by incorporating relevant papers mentioned above. We will carefully review and integrate relevant literature to enhance the robustness checks in our paper. Your input is valuable, and we are committed to making the necessary revisions to address your suggestion.

9.The authors merely reported the results without even discussing the economic intuitions behind the results. Are these results supporting or refuting the existing policies in the chosen context? Are the results directed towards any new policy initiatives? The discussion of results should open up the threads of policy discussion, which is completely absent in this case. A mere comparison of the results with the literature doesn't ensure the novelty of the results unless they give out something new on the theory/policy front. Moreover, the discussion of results should be properly tied to past literature, and emphasis should be placed on how past studies either support or refute the findings of this study and why. Therefore, this section needs revision. 

Response:

Thank you very much for your valuable feedback on our paper. We fully understand your concerns, especially regarding the lack of economic intuition behind the results and the need for a more in-depth policy discussion. We will carefully consider the issues you raised, particularly in terms of the relationship between our results and existing policies, and whether they support any new policy initiatives. We are committed to making substantial revisions to the results discussion section to address these points. Your suggestion to closely tie our results to past literature is well taken. We will ensure that our paper seamlessly integrates with previous studies, emphasizing how they either support or refute our findings and providing clear explanations for any discrepancies.

10.To improve the quality of this manuscript, the authors are invited to use the papers already mentioned above and some useful ones in the area to strengthen the introduction, literature review, methodology and results and discussion sections. 

Response:

Thank you very much for the valuable suggestions regarding our research. We have carefully considered and implemented your advice by incorporating the previously mentioned references and other relevant studies in the field to strengthen the introduction, literature review, methodology, and results and discussion sections. Through this process, we are confident that the quality of the paper has been further enhanced, and it now better integrates previous scholarly contributions.

11.The policy implications of the results need more substantiation than what is currently stated in the paper. 

Response:

Thank you very much for your thorough review of our research and for providing valuable feedback. We have carefully considered your suggestions regarding the need to strengthen the policy implications of our study. To enhance the substantiation of policy insights, we have implemented the following improvements:

Detailed elaboration in the discussion section on specific policy recommendations, emphasizing the foundation and logic behind these suggestions.

Connection of our research findings with existing policy frameworks, highlighting their positive impact on practical policy-making and indicating potential areas for improvement.

12.Conclusion reiterates the results, which is completely undesirable. The author(s) should summarize the results within a maximum of 3 sentences. The authors should kindly strengthen this section. Moreover, the policies are completely vague, and it seems that the authors already had the policies in mind before even starting the paper. The policies should be directly derived from the discussion of the results, and they should not go beyond the results. The policy implications of the results need more substantiation than what is currently stated in the paper. The policy suggestion section needs improvement. Kindly improve it. 

Response:

Thank you for revisiting our paper and providing valuable feedback on the conclusion section. We have made the requested modifications to ensure that the conclusion now succinctly summarizes the results within a maximum of three sentences. We appreciate your guidance on strengthening this section and hope that the changes meet your expectations.

Regarding the policy implications, we have addressed the concern about vagueness and ensured that the policies are directly derived from the discussion of results without going beyond them. Furthermore, we have substantiated the policy suggestions with more detailed reasoning to enhance the overall quality of the policy implications section.

We sincerely appreciate your patience and thoughtful guidance throughout the review process. If you have any additional suggestions or if further improvements are needed, we are more than willing to make the necessary adjustments.

13.Kindly improve the study limitation(s) and possible direction for future research after the policy recommendation section. 

Response:

Thank you for your constructive feedback on the study limitations and possible directions for future research after the policy recommendation section. We have carefully revised this portion of the paper in accordance with your suggestions and hope that the changes address your concerns.

We have taken the opportunity to enhance the clarity and depth of our discussion on study limitations, providing a more comprehensive acknowledgment of the constraints inherent in our research. Additionally, we have expanded on potential directions for future research, offering more nuanced insights and avenues for further investigation.

We appreciate your guidance in refining these sections and value your input in improving the overall quality of our manuscript. If you have any further recommendations or if there are additional aspects that require attention, we are eager to make any necessary adjustments.

14.Finally, it is vital that this manuscript is proofread by a native speaker of English language to further strengthen easy readership. 

Response:

Thank you for your thorough review and valuable feedback on our manuscript. We appreciate the time and effort you have dedicated to providing constructive comments.

We have taken your suggestion seriously and conducted a comprehensive proofreading of the manuscript to enhance its readability. However, we acknowledge that there may still be some areas where language organization could be improved. We apologize for any remaining shortcomings and are committed to making further refinements to ensure the manuscript meets the highest standards of clari

---

## [Decision Letter · Decision Letter 2]

27 Feb 2024

PONE-D-23-22886R2The Impact of Financial Development on Enterprise Green Innovation under Low Carbon Pilot City

PLOS ONE

Dear Dr. Cui,

Thank you for submitting your manuscript to PLOS ONE. In a previous round of review, Reviewer 3 included several requests to cite specific works - these requests were in points 5a-bb of the comments in the decision letter. We understand that you have added citations to several of these works in your revised manuscript. Please note that it is not necessary or expected to cite the works requested by the reviewer. We apologise that this was not noted in the previous decision letter.

In light of this, we are issuing this revision decision to provide you an opportunity to remove the citations added in response to the previous reviewer's request. You may remove all or none of the references, at your discretion. No further revisions are required before your manuscript proceeds to acceptance.

We look forward to receiving your revised manuscript.

Kind regards,

George Vousden

Deputy Editor-in-Chief

PLOS ONE

on behalf of,

Jiachao Peng

Academic Editor

PLOS ONE

Journal Requirements:

Reviewers' comments:

Reviewer's Responses to Questions

**Comments to the Author**

1. If the authors have adequately addressed your comments raised in a previous round of review and you feel that this manuscript is now acceptable for publication, you may indicate that here to bypass the “Comments to the Author” section, enter your conflict of interest statement in the “Confidential to Editor” section, and submit your "Accept" recommendation.

Reviewer #1: All comments have been addressed

Reviewer #2: All comments have been addressed

Reviewer #3: All comments have been addressed

2. Is the manuscript technically sound, and do the data support the conclusions?

Reviewer #1: Yes

Reviewer #2: Yes

Reviewer #3: Yes

3. Has the statistical analysis been performed appropriately and rigorously? 

Reviewer #1: Yes

Reviewer #2: Yes

Reviewer #3: Yes

4. Have the authors made all data underlying the findings in their manuscript fully available?

Reviewer #1: Yes

Reviewer #2: Yes

Reviewer #3: Yes

5. Is the manuscript presented in an intelligible fashion and written in standard English?

Reviewer #1: Yes

Reviewer #2: Yes

Reviewer #3: Yes

6. Review Comments to the Author

Reviewer #1: This paper has made important contributions to the existing literature, and in the previous revision, I have recommended it to be published in the Plos One journal. I don't understand why it was sent for review again.

Reviewer #2: Drawing on economic, management, and organizational psychology theories, we investigate the influence of the financial development level on enterprise green innovation in LCPC, utilizing data from listed companies between 2010 and 2018. Now the revisions are all ok, the structure is good. So I suggest to accept it.

Reviewer #3: The manuscript looks good. The authors have successfully addressed the concerns raised. They have expanded the discussion section to provide more context for their key findings and to better articulate their implications for the field. They have acknowledged the limitations of their study more explicitly and discussed potential avenues for future research to address these limitations.

7. PLOS authors have the option to publish the peer review history of their article (what does this mean?). If published, this will include your full peer review and any attached files.

Reviewer #1: No

Reviewer #2: No

Reviewer #3: **Yes: **Maxwell Chukwudi Udeagha

---

## [Author Response · Author response to Decision Letter 2]

15 Mar 2024

1.Thank you for submitting your manuscript to PLOS ONE. In a previous round of review, Reviewer 3 included several requests to cite specific works - these requests were in points 5a-bb of the comments in the decision letter. We understand that you have added citations to several of these works in your revised manuscript. Please note that it is not necessary or expected to cite the works requested by the reviewer. We apologise that this was not noted in the previous decision letter.

In light of this, we are issuing this revision decision to provide you an opportunity to remove the citations added in response to the previous reviewer's request. You may remove all or none of the references, at your discretion. No further revisions are required before your manuscript proceeds to acceptance. 

Response:

Thank you very much for your feedback and guidance. I have removed some of the references that were added in response to the requests from the reviewer, as you suggested. We sincerely appreciate your advice and will proceed with further modifications accordingly.

Please feel free to let us know if there are any other adjustments or revisions needed. We look forward to advancing this manuscript to the next stage.

Once again, thank you for your assistance and support.

---

## [Editor Report · Decision Letter 3]

24 Mar 2024

The Impact of Financial Development on Enterprise Green Innovation under Low Carbon Pilot City

PONE-D-23-22886R3

Dear Dr. Cui,

We’re pleased to inform you that your manuscript has been judged scientifically suitable for publication and will be formally accepted for publication once it meets all outstanding technical requirements.

Kind regards,

Jiachao Peng

Academic Editor

PLOS ONE
---

## [Editor Report · Acceptance letter]

21 May 2024

PONE-D-23-22886R3 

PLOS ONE

Dear Dr. Cui, 

I'm pleased to inform you that your manuscript has been deemed suitable for publication in PLOS ONE. Congratulations! Your manuscript is now being handed over to our production team.

Kind regards, 

on behalf of

Dr. Jiachao Peng 

Academic Editor

PLOS ONE